# NaCl Modifies Biochemical Traits in Bacterial Endophytes Isolated from Halophytes: Towards Salinity Stress Mitigation Using Consortia

**DOI:** 10.3390/plants13121626

**Published:** 2024-06-12

**Authors:** Jesús Adrián Barajas González, Yersaín Ely Keller de la Rosa, Rogelio Carrillo-González, Ma. del Carmen Ángeles González-Chávez, María Eugenia Hidalgo Lara, Ramón Marcos Soto Hernández, Braulio Edgar Herrera Cabrera

**Affiliations:** 1Programa en Edafología, Colegio de Postgraduados, Campus Montecillo, Carr. México-Texcoco km 36.5, Montecillo 56230, Mexico; barajas.jesus@colpos.mx (J.A.B.G.); crogelio@colpos.mx (R.C.-G.); 2Departamento de Biotecnología y Bioingeniería, CINVESTAV, Av. IPN 2508, Ciudad de México 07360, Mexico; yersainkeller@ciencias.unam.mx (Y.E.K.d.l.R.); ehidalgo@cinvestav.mx (M.E.H.L.); 3Programa en Botánica, Colegio de Postgraduados, Campus Montecillo, Carr. México-Texcoco km 36.5, Montecillo 56230, Mexico; msoto@colpos.mx; 4Programa en Estrategias de Desarrollo Agrícola Regional, Colegio de Postgraduados, Campus Puebla, Carr. Fed. Mex-Pue, Puebla 72130, Mexico; behc@colpos.mx

**Keywords:** plant growth-promoting bacteria, solubilization, salinity, enzyme activity, organic acids, food safety, plant fitness

## Abstract

Bacterial endophytes (120) were isolated from six halophytes (*Distichlis spicata*, *Cynodon dactylon*, *Eragrostis obtusiflora*, *Suaeda torreyana*, *Kochia scoparia*, and *Baccharis salicifolia*). These halophiles were molecularly identified and characterized with or without NaCl conditions. Characterization was based on tests such as indole acetic acid (IAA), exopolysaccharides (EPS), and siderophores (SID) production; solubilization of phosphate (P), potassium (K), zinc (Zn), and manganese (Mn); mineralization of phytate; enzymatic activity (acid and alkaline phosphatase, phytases, xylanases, and chitinases) and the mineralization/solubilization mechanisms involved (organic acids and sugars). Moreover, compatibility among bacteria was assessed. Eleven halophiles were characterized as highly tolerant to NaCl (2.5 M). The bacteria isolated were all different from each other. Two belonged to *Bacillus velezensis* and one to *B. pumilus* while the rest of bacteria were identified up to the genus level as belonging to *Bacillus*, *Halobacillus, Halomonas, Pseudomonas, Nesterenkonia*, and three strains of *Oceanobacillus.* The biochemical responses of nutrient solubilization and enzymatic activity were different between bacteria and were influenced by the presence of NaCl. Organic acids were involved in P mineralization and nutrient solubilization. Tartaric acid was common in the solubilization of P, Zn, and K. Maleic and vanillic acid were only detected in Zn and K solubilization, respectively. Furthermore, sugars appeared to be involved in the solubilization of nutrients; fructose was detected in the solubilization tests. Therefore, these biochemical bacterial characteristics should be corroborated in vivo and tested as a consortium to mitigate saline stress in glycophytes under a global climate change scheme that threatens to exacerbate soil salinity.

## 1. Introduction

Salinity affects 20% of the world’s agricultural soils and is one of the leading causes of yield reduction in crops of economic interest, most of which are sensitive to salinity [1,2]. Soil salinity and sodicity will increase considerably due to climate change; therefore, they have become a problem of global concern. Salt stress decreases the yield, nutrient content, and nutraceutical quality of food from crops of economic interest, which can cause deficiencies in humans and favor hidden hunger. Halophyte plants are essential for creating saline ecosystems due to their high tolerance and remediation potential [3]. These plants have evolved several strategies to survive in these environments, including increased cytoplasmic osmotic pressure, the production of compatible solutes, and the exclusion of sodium ions from the cells, or their accumulation in the vacuole [4]. Another alternative used by halophyte plants to attenuate the effect of salt stress is related to their microbiome.

For decades, most research focused on rhizobacteria from saline soils and isolated from glycophytic crops (salinity-sensitive plants) [5,6]. However, more recently, research has focused on endophytic bacteria and their isolation from halophytes. Conventionally, endophytes are defined as bacteria or fungi that reside intercellularly in plant tissues and do not cause adverse effects on plant growth [7]. Endophytes have advantages over rhizospheric ones because they are protected from biotic and abiotic environmental challenges. Therefore, endophytes are a potential tool for improving plant growth and yield under salt-stress conditions. Studies that have isolated endophytic bacteria from halophytes suggest increased adaptation and survival of these bacteria in saline. Moreover, inoculation with endophytic and high-salinity-tolerant (halophilic) bacteria increases the tolerance of plants grown in saline soils [8].

Halophilic endophytic bacteria may have similar properties to rhizobacteria. For example, they solubilize nutrients [such as phosphorous (P), potassium (K), zinc (Zn) or manganese (Mn)] to improve the efficiency of nutrient uptake, fix atmospheric nitrogen (N_2_), produce exopolysaccharides (EPS) and siderophores [iron (Fe) chelation], and modulate the level of phytohormones within plant tissues [9]. Endophytic bacteria can also synthesize plant hormones (auxins, cytokinins, abscisic acid, and gibberellins) that modify plant physiology to resist stress conditions [10]. 

Salinity is known to interfere with the availability of Fe, N, and other essential plant elements [11]. Endophytic bacteria can also secrete siderophores, high-affinity Fe-chelating compounds, so that plants can easily access Fe-siderophore complexes for nutrition [9]. Some endophytic bacteria are N_2_-fixers and represent an important source of available N in saline-sodic soils. Phosphorus is an essential plant element, but its availability is commonly low in saline soils. P-solubilizing and -mineralizing bacteria improve P nutrition in plants through chelation, ion exchange, acidification (by secretion of low molecular weight organic acids), and enzyme production (phosphatases and phytases) [12,13,14]. However, salinity is one of the main factors negatively influencing P solubilization by bacteria [15]. 

In saline–sodic soils, pH can decrease the availability of elements that play crucial metabolic roles in plants, such as micronutrients. These micronutrients act as cofactors for several enzymes that are vital for plant development [16]. Their deficiencies can significantly impact crops of economic interest [17]. As mentioned, bacteria can potentially solubilize other nutrients like K, Zn, or Mn. However, the processes and mechanisms involved in this solubilization remain poorly understood. 

Endophytic bacteria are known for their production of various enzymes, including hydrolytic proteases, cellulases, hemicellulases, xylanases, chitinases, pectinases, glucanases, and pectinases [9]. Some of these extracellular enzymes assist plants in establishing systemic resistance against invasion by phytopathogenic bacteria or fungi. This information highlights the need to inoculate glycophytic crops with microorganisms from saline soils or halophytes. However, there is limited information on the effect of the presence or absence of NaCl on the plant growth-promoting properties of halophilic endophytic bacteria. Similarly, there is a dearth of information on compatible and functional bacterial consortia that can effectively promote plant growth and mitigate salt stress, highlighting the need for further investigation in this area.

Utilizing microbial consortia as an inoculum, particularly those derived from extreme environments, can be an effective biotechnology tool [18,19]. These consortia, complementing each other, have the potential to enhance plant growth [20]. The research by Vaishnav et al. [21] underscores the importance of harnessing bacterial inoculants from saline environments for sustainable agriculture, addressing plant salt stress. Marghoob et al. [22] and Rajput et al. [23] have demonstrated the effectiveness of a consortia of *Aeromonas* sp. and *A*. *salmonicida*, or *Pantoea* sp. and *Erwinia rhapontici*, in promoting wheat growth. Similarly, the bacterial consortia of *Lysinibacillus* sp. and *Bacillus* sp. encouraged tomato growth [24]. Therefore, consortia-based bacterial inoculant production from halophytes may be useful for salt-sensitive plants.

The objectives of the present research were as follows: (1) to isolate endophytic bacteria from the roots of seven halophytes established in saline soils; (2) to molecularly identify endophytic bacteria selected for their halotolerance; (3) to biochemically characterize these halophile bacteria selected under two salinity conditions (0 M and 2.5 M NaCl) for potential use in plant growth promotion; (4) to determine the production of organic acids and sugars in selected halophilic bacteria in P, Zn, K, and Mn solubilization; (5) to evaluate the activity of four extracellular enzymes in the halophilic endophytic bacteria; and (6) to determine bacterial compatibility to form consortia for future use in salinity mitigation in glycophytic plants.

## 2. Results 

### 2.1. Isolation and Molecular Identification of Endophytic Bacteria by Halotolerance

A total of 120 endophytic bacterial isolates were obtained from halophyte roots. Fifty-eight endophyte isolates were tolerant to 0.1 M NaCl, 28 to 0.5 M NaCl, 22 to 1.5 M NaCl, and 11 to 2.5 M NaCl. The latter 11 halophilic isolates were molecularly identified, and phylogenetic analysis of the DNA sequences generated by 16S rRNA gene-sequencing using NCBI databases revealed that each isolate corresponded to a single species, except isolates 4 and 7 (Figure 1). The BLAST algorithm for the sequences showed that isolate 1 corresponds to *Oceanobacillus* sp., isolate 2 to *Bacillus* sp., isolate 3 to *Nesterenkonia* sp., isolate 4 to *Bacillus velezensis*, isolate 5 to *Halobacillus* sp., isolate 6 to *Oceanobacillus* sp., isolate 7 to *Bacillus velezensis*, isolate 8 to *Halomonas* sp., isolate 9 to *Bacillus pumilus*, isolate 10 to *Pseudomonas* sp., and isolate 11 to *Oceanobacillus* sp., with degrees of proximity of 99% (Appendix A). This is the first report of *B. velezensis* and *B*. *pumilus* as endophytes isolated from halophytes.

### 2.2. Indolacetic Acid and Siderophore Production

All the tested bacterial isolates produced IAA. Bacteria growing in 2.5 M NaCl-spiked media showed a significant increase in IAA production (*p* ≤ 0.001), except in isolates 10 (*Pseudomonas* sp.) and 11 (*Oceanobacillus* sp.) (Figure 2a). The maximum concentration of IAA, with 2.5 M NaCl, was observed in *Bacillus* sp. (isolate 2) and *B. velezensis* (isolate 7), with 19.9 mg L^−1^ and 20.5 mg L^−1^, respectively. Without NaCl, the maximum concentration (12.8 mg L^−1^) was observed with *Oceanobacillus* sp. (isolate 11). 

Isolates 1 (*Oceanobacillus* sp.), 2 (*Bacillus* sp.), 6 (*Oceanobacillus* sp.), 7 (*B. velezensis*), and 10 (*Pseudomonas* sp.) increased siderophore production in the saline medium compared to that of the control (Figure 2b). In the rest of the bacteria, the percentage of siderophores was similar with and without NaCl. Isolate 8 (*Halomonas* sp.) produced the highest siderophores percentage (71–73%) in both salinity conditions. 

### 2.3. Nitrogenase Enzyme Activity and Exopolysaccharide Production

Isolate 3 (*Nesterenkonia* sp.), 4 (*B. velezensis*), 8 (*Halomonas* sp.), 7 (*B. velezensis*), and 9 (*B. pumilus*) had diazotrophic activity in the absence of NaCl (Appendix A), which means that 45% of the bacteria presented this characteristic. In the presence of 2.5 M NaCl, isolates 3, 4, 8, 7, 9, and 10 showed diazotrophic activity. 

This study is among the pioneering works exploring EPS production in halophytic endophytes. Only 4 of 11 halophilic endophytic bacterial isolates were positive for EPS production (1, 4, 5, and 6) (Appendix A). Remarkably, only isolate 4 (*B. velezensis*) was positive for both EPS production and N_2_ fixation.

### 2.4. Nutrient Solubilization, Bacterial Growth, and pH Modification 

#### 2.4.1. Inorganic and Organic P

The ability of bacteria to solubilize P from Ca_3_(PO_4_)_2_ was decreased by the presence of NaCl (Figure 3a). The highest P solubilization in the absence of NaCl was observed for isolates 3 (*Nesterenkonia* sp.), 5 (*Halobacillus* sp.), 6 (*Oceanobacillus* sp.), and 10 (*Pseudomonas* sp.), with 15.5 mg P L^−1^, 15.9 mg L^−1^, 15.8 mg L^−1^, and 15.8 mg L^−1^, respectively. The maximum P solubilization was 11.1 mg L^−1^ in isolate ten, at 2.5 M NaCl. 

In the P solubilization test, the bacteria not only solubilized P but also acidified the culture medium. The average pH of the uninoculated NBRIP medium was 6.8 with and 6.9 without NaCl (Figure 3b). After 48 h of incubation, the average pH of the cultures was pH = 5.4 without NaCl and pH = 5.9 with NaCl. This acidification process was found to be partially responsible for P solubilization, as evidenced by positive correlations between soluble P concentration and pH with (r = 0.75) and without NaCl (r = 0.91). The lowest pH in the medium was observed with isolate 10 (*Pseudomonas* sp.) at both NaCl concentrations (pH = 4.3 in the absence of NaCl and pH = 4.6 with 2.4 M NaCl). This strain showed one of the highest P solubilization values (Figure 3a). 

In all solubilization tests, no differences in biomass were obtained among bacteria in the absence or presence of NaCl (Appendix A). Moreover, the presence or absence of NaCl did not influence bacterial growth in all solubilization tests. Overall, bacterial biomass did not correlate with the solubilization capacity of the different studied elements.

This research shows the first report on the influence of NaCl on the production and concentration of organic acids of halophilic endophytic bacteria that solubilize inorganic P (Table 1). All bacteria secreted citric and succinic acids in the presence and absence of NaCl. Isolate 4 (*B. velezensis*) produced the highest concentration of citric acid, 15.5 mg L^−1^ with 2.5 M NaCl and 15.2 mg L^−1^ without NaCl. At the same time, no differences were observed among bacteria in succinic acid production in both salt conditions. In the absence of NaCl, all bacteria produced lactic acid, whereas in the presence of salt, no acid was produced by isolates 3 (*Nesterenkonia* sp.), 8 (*Halomonas* sp.), and 9 (*B. pumilus*). In the presence of NaCl, *Oceanobacillus* sp. isolates (1 and 10) produced the highest concentration of lactic acid (Table 1) compared to the other bacteria. NaCl increased lactic acid concentration only in isolate 6 (*Oceanobacillus* sp.), 7 (*B. velezensis*), 8 (*Halomonas* sp.), 9 (*B. pumilus*), and 10 (*Pseudomonas* sp.). Tartaric acid production was detected in isolates 3 (*Nesterenkonia* sp.) and 9 (*B. pumilus*), both in the absence and presence of NaCl, and no differences were observed in the tartaric acid production among these bacteria. Isolate 1 (*Oceanobacillus* sp.), 2 (*Bacillus* sp.), and 8 (*Halomonas* sp.) secreted higher amounts of citric acid in the presence of NaCl. However, NaCl did not increase the concentration of succinic acid, except in isolate 8 (*Halomonas* sp.). 

This study is also the first to evaluate the mineralization of organic P from phytic acid in the absence and presence of NaCl. Like the phosphate solubilization from Ca_3_(PO_4_)_2_, organic P mineralization was reduced in the presence of NaCl in all bacteria except isolates 1 (*Oceanobacillus* sp.) and 4 (*B. velezensis*). Bacterial isolates produced higher phosphate concentrations when phytic acid was used than when Ca_3_(PO_4_)_2_ was used (Figure 3c). Without NaCl, the average soluble phosphate concentration from phytic acid was 75% higher than with Ca_3_(PO_4_)_2_. Isolates 3 (*Nesterenkonia* sp.) and 9 (*B. pumilus*) showed the highest organic P mineralization in the presence of NaCl, but in the absence of NaCl, isolate 5 showed the highest (*Halobacillus* sp.) (Figure 3c). In the present study, pH was not significantly modified during P mineralization with and without NaCl, contrary to what was observed with inorganic P. The average pH in the cultures without NaCl was 6.7, while in the presence of NaCl, the average pH was 6.8. The original pH of the Luria Bertani (LB) broth without NaCl was 7.0, and with NaCl, it was 7.3 (Figure 3d). 

#### 2.4.2. Solubilization of K, Mn, and Zn

Isolates 2 (*Bacillus* sp.), 3 (*Nesterenkonia* sp.), 4 (*B. velezensis*), 5 (*Halobacillus* sp.), 6 (*Oceanobacillus* sp.), and 8 (*Halomonas* sp.) increased their ability to solubilize K in the presence of NaCl. However, in the other isolates, solubilization was independent of NaCl. Isolate 4 was the most efficient in K solubilization (57.1 mg L^−1^) with salt, therefore, it could be used to improve the availability of this element in salinity-sensitive plants. In the non-saline medium, isolate 7 (*B. velezensis*) had the highest K solubilization (14.5 mg L^−1^, Figure 4a). The present investigation found a correlation between pH and K solubilization in both salinity conditions (r = 0.74).

All bacterial isolates diminished the pH of the culture medium (Figure 4b), but *B. velezensis* (4) reduced it more significantly, to 5.1. In addition, all bacteria produced tartaric, malic, and citric acids, except isolate 8 (*Pseudomonas* sp.), which also produced vanillic acid, which in this study, is related, for the first time, to K solubilization. The isolates did not produce fumaric, oxalic, maleic, salicylic, succinic, lactic, and acetic acids (Table 2). Isolate 4 (*B. velezensis*) had the highest concentration of tartaric acid in saline conditions (10.8 mg L^−1^), while in a non-saline medium, isolates 7 (*B. velezensis*) and 3 (*Nesterenkonia* sp.) did, with 8.8 mg L^−1^. Isolate 9 (*B. pumilus*) produced the highest concentration of malic acid with and without NaCl (8.7 mg L^−1^ and 7.6 mg L^−1^, respectively); it also had the highest concentration of citric acid (Table 2). Salinity influenced the concentration of some organic acids, and this effect was different among isolates. Isolates 1 (*Oceanobacillus* sp.), 7 (*B. velezensis*), and 10 (*Pseudomonas* sp.) produced more citric acid in the absence of salt. In contrast, isolates 2 (*Bacillus* sp.), 3 (*Nesterenkonia* sp.), 9 (*B. pumilus*), and 10 (*Pseudomonas* sp.) produced more malic acid in the presence of NaCl. Similarly, tartaric acid increased in the presence of salt in isolate 1 (*Oceanobacillus* sp.), 4 (*B. velezensis*), 5 (*Halobacillus* sp.), 6 (*Oceanobacillus* sp.), 8 (*Halomonas* sp.), 10 (*Pseudomonas* sp.), and 11 (*Oceanobacillus* sp.). However, in isolate 3 (*Nesterenkonia* sp.) and 7 (*B. velezensis*) the concentration of tartaric acid decreased in the presence of salt. 

All isolates were able to solubilize Mn. Adding NaCl to the medium only decreased Mn solubilization by *Halomonas* sp., *Nesterenkonia* sp., *Pseudomonas* sp., and *Oceanobacillus* sp., but not by the other bacteria (Figure 4c). Isolate 5 (*Halobacillus* sp.) solubilized the highest concentration of Mn, similarly with and without salt (1.3 mg L^−1^). 

All endophytic bacteria during Mn solubilization acidified the culture medium under both salinity conditions (Figure 4b,d,f). However, unlike P, K, and Zn solubilization, no correlation was observed between pH and Mn solubilization (r = 0.21) in either salinity condition. 

Only malic and citric acids were detected in Mn solubilization (Table 2). None of the isolates produced fumaric, oxalic, lactic, acetic, succinic, tartaric, vanillic, or salicylic acids. The influence of NaCl on the production of these two organic acids depended on the bacterial isolates. Only isolates 1 (*Oceanobacillus* sp.), 2 (*Bacillus* sp.), 3 (*Nesterenkonia* sp.), and 9 (*B. pumilus*) produced malic acid in the absence and presence of NaCl. In the absence of NaCl, isolate 9 (*B. pumilus*) produced the highest concentration of malic acid (7.8 mg L^−1^), whereas, in the presence of NaCl, isolates 1 (*Oceanobacillus* sp.) and 2 (*Bacillus* sp.) produced the highest concentration of this acid (7.4 and 7.2 mg L^−1^, respectively). The presence of NaCl only increased citric acid concentrations in isolates 1 (*Oceanobacillus* sp.), 2 (*Bacillus* sp.), and 8 (*Halomonas* sp.), which showed similar values. In contrast, isolates 1, 2, and 9 increased their malic acid production in the presence of NaCl.

All isolates solubilize ZnO; however, it showed differences depending on the presence of salt and the bacterial strain (Figure 4e). Isolate 8 (*Halomonas* sp.) solubilized more Zn (13.3 mg L^−1^) in the presence of NaCl than the other isolates. In the absence of NaCl, isolate 4 (*B. velezensis*) had a higher solubilization capacity (107 mg L^−1^) than the other bacteria. Regarding the effect of NaCl, isolates 2 (*Bacillus* sp.), 4 (*B. velezensis*), 5 (*Halobacillus* sp.), and 11 (*Oceanobacillus* sp.) solubilized more Zn in the absence of NaCl than in 2.5 M NaCl. All bacteria in both NaCl conditions produced maleic and malic acids. In contrast with P, K, and Mn solubilization, only bacterial strains 5 (*Halobacillus* sp.), 8 (*Halomonas* sp.), and 10 (*Pseudomonas* sp.) produced citric acid upon Zn solubilization at both NaCl conditions (Table 2). Isolates 1 (*Oceanobacillus* sp.), 2 (*Bacillus* sp.), 3 (*Nesterenkonia* sp.), 7 (*B. velezensis*), 8 (*Halomonas* sp.) and 11 (*Oceanobacillus* sp.) were the only ones that secreted tartaric acid regardless of NaCl concentrations.

No isolate produced fumaric, lactic, acetic, succinic, ferulic, or salicylic acid (Table 2). Isolates 4 (*B. velezensis*), 8 (*Halomonas* sp.), and 9 (*B. pumilus*) secreted the highest concentration of maleic acid in the presence of NaCl in Zn solubilization. In contrast, without NaCl, the highest concentration was secreted by isolate 4 (11.8 mg L^−1^). Isolate 8 produced the highest concentration of tartaric acid, 11.6 mg L^−1^ with salt and 11.8 mg L^−1^ without salt. The present work is the first to identify malic acid production in Zn solubilization. In bacteria that secreted citric and tartaric acid, there was no difference in the concentration of these acids in the presence and absence of salt. In contrast, the presence of salt increased the concentration of malic acid in isolates 2 (*Bacillus* sp.), 3 (*Nesterenkonia* sp.), 4 (*B. velezensis*), 6 (*Oceanobacillus* sp.), and 7 (*B. velezensis*). The concentration of maleic acid changed in the presence of salt in isolates 1 (*Oceanobacillus* sp.), 5 (*Halobacillus* sp.), 7 (*B*. *velezensis*), 8 (*Halomonas* sp.), 9 (*B. pumilus*), and 10 (*Pseudomonas* sp.) compared to the absence of salt.

#### 2.4.3. Interaction between Organic Acids and the Solubilization Ion

Some organic acids produced by the bacteria were common in the solubilization of different ions (Appendix A). In the solubilization of P, Mn, and K, citric acid was common and produced by all bacteria regardless of the presence of NaCl. In Zn solubilization, citric acid was detected only in four isolates. Succinic and lactic acids were only identified in the inorganic P solubilization. Tartaric acid was observed in the solubilization of P, Zn, and K, but not Mn. Malic acid was detected in the solubilization of Zn, Mn, and K, but not P. Maleic acid was only observed in the solubilization of Zn. Vanillic acid was only produced by isolate 8 (*Halomonas* sp.) during K solubilization. 

#### 2.4.4. Production of Sugars in Nutrient Solubilization Tests

For the first time, this research identified sugars present in the solubilization media of different nutrients (inorganic and organic P, K, Zn, and Mn) by the influence of NaCl. The presence or absence of salt impacted fructose concentrations, depending on the solubilization ion and the bacterial strain (Table 3). Fructose production in Ca_3_(PO_4_)_2_ solubilization was common, except in isolate 5. In contrast, xylose production depended on the bacterial strain. None of the isolates produced arabinose, trehalose, maltose, sucrose, or lactose. Glucose was detected in the solubilization extracts but not reported since it was used in the culture broths as a carbon source. The highest average concentration of fructose was observed in inorganic P solubilization compared to the other solubilization ions, both in the absence and presence of NaCl. During the solubilization of inorganic P without NaCl, the average fructose concentration was 166.4 µg µL^−1^_,_ and it is 1.2-fold higher (208.4 µg µL^−1^) in the condition with NaCl. The two isolates of *B. velezensis* (isolates 4 and 7) produced the highest concentration of fructose both in the absence (458.8 and 463.8 µg µL^−1^, respectively) and presence of NaCl (459.4 and 456.2 µg µL^−1^, respectively). Except for isolates 4 and 7, fructose concentration increased under salt conditions (Table 3).

In organic P mineralization, the fructose concentration was 13.6 µg µL^−1^ without salt and 14.5 µg µL^−1^ with salt. Fructose produced in the solubilization of inorganic P in the absence of salt was seven times higher than in organic P and ten times higher than in the presence of salt. In organic P solubilization, no differences in fructose concentration were observed due to the effect of salt. Isolate 6 (*Oceanobacillus* sp.) produced high concentrations with and without salt (95.1 and 93.1 µg µL^−1^, respectively). Similar to the solubilization of inorganic P, isolates 4 and 7 (*B. velezensis*) produced the most fructose in K, Mn, and Zn solubilization. The concentration of fructose in K solubilization was similar in both bacteria with (267.6 and 264.8 µg µL^−1^) and without salt (157.5 to 153.1 µg µL^−1^). Zn solubilization without salt (417.9 and 414.0 µg µL^−1^) was higher than when salt was added (349.4 and 347.4 µg µL^−1^) in the same two bacteria, respectively. Fructose was similarly produced in the absence or presence of NaCl (16.0 and 16.5 µg µL^−1^, respectively) during Mn solubilization. Moreover, isolates 4 and 7 secreted the highest concentration of fructose, both in the presence (33.4 and 32.0 µg µL^−1^, respectively) and absence of salt (32.0 and 35.0 µg µL^−1^, respectively). The concentration of fructose in the solubilization of inorganic P increased in the presence of salt in all bacteria. In contrast, the presence or absence of salt did not influence the concentration of fructose coming from the extract of the mineralization of organic P. Regarding K solubilization, all bacteria increased fructose secretion in the presence of salt, except isolates 3 (*Nesterenkonia* sp.) and 11 (*Pseudomonas* sp.). In Mn solubilization, only isolate 9 (*B. pumilus*) and 10 (*Pseudomonas* sp.) increased fructose concentration in the presence of salt, while the other bacteria had similar fructose concentrations. NaCl increased fructose concentration in the Zn solubilization extracts only in isolates 6 (*Oceanobacillus* sp.) and 11 (*Oceanobacillus* sp.; Table 3).

In inorganic P solubilization, xylose was only produced by isolates 10 (*Pseudomonas* sp., with 229.5 µg µL^−1^) and 5 (*Halobacillus* sp. with 29.0 µg µL^−1^). During P mineralization with and without salt, isolates 3 (*Nesterenkonia* sp.), 4 (*B. velezensis*), 5 (*Halobacillus* sp.), and 7 (*B. velezensis*) produced xylose. *Nesterenkonia* sp. produced the highest concentration of xylose with 2.5 (8.0 µg µL^−1^) and 0 M NaCl (8.5 µg µL^−1^). In K solubilization, the highest xylose concentration was observed in isolates 3 (*Nesterenkonia* sp.), 4 (*B. velezensis*), 7 (*B. velezensis*), and 9 (*B. pumilus*). In Mn solubilization, isolates 4 and 7 had the highest concentrations; in Zn solubilization, isolates 3 and 9 had more xylose (Table 3). In K solubilization, NaCl did not influence xylose concentrations in isolates 3 (*Nesterenkonia* sp.), 4 (*B. velezensis*), 7 (*B. velezensis*), and 9 (*B. pumilus*); the same result was observed for Mn in isolates 4 and 7 (*B. velezensis*), and for Zn in isolates 3 (*Nesterenkonia* sp.) and 9 (*B. pumilus*). Contrary to what was observed for fructose concentration, adding NaCl influenced only the xylose concentration in the broth during the solubilization of inorganic P in isolates 5 (*Halobacillus* sp.) and 10 (*Pseudomonas* sp.).

### 2.5. Enzyme Activity in Different Salinity Conditions

All 11 bacteria showed alkaline and acid phosphatase enzyme activity (Table 4), but NaCl decreased alkaline phosphatase enzyme activity, whereas acid phosphatase activity increased. The maximum alkaline phosphatase activity was observed without NaCl in all bacteria except isolate 8 (*Halomonas* sp.), which only showed activity in the presence of salt (Table 4). The highest alkaline phosphatase activity (34.5 µg mL^−1^ h^−1^) in 0 M NaCl was observed in isolate 2 (*Bacillus* sp.). At 1.5 M NaCl, all bacteria had a similar alkaline phosphatase activity, except for isolate 7 (*B. velezensis*). 

The average maximum acid phosphatase activity in all bacteria was observed at 1.5 M NaCl (58.7 µg mL^−1^ h^−1^); this was 2.6 times higher than that for the average alkaline phosphatase activity (22.5 µg mL^−1^ h^−1^) at 0 M NaCl. The average activity of the two phosphatase enzymes at 0 M NaCl was similar; however, when comparing these at 1.5 and 2.5 M NaCl, acid phosphatase was 7.4 and 8 times higher than the alkaline one. At 0 M NaCl, the highest enzymatic activity for acid phosphatase (51.3 µg mL^−1^ h^−1^) was found in isolate 10 (*Pseudomonas* sp.). In contrast, isolates 1 (*Oceanobacillus* sp.), 2 (*Bacillus* sp.), 3 (*Nesterenkonia* sp.), and 8 (*Halomonas* sp.) showed no activity. At 1.5 M NaCl, eight isolates had activity higher than the average one (58.7 µg mL^−1^ h^−1^). At 2.5 M NaCl, isolates 4 (*B*. *velezensis*) and 5 (*Halobacillus* sp.) had the highest activity.

All 11 halophilic endophytic bacteria produced phytase (Table 4) and released soluble P to use phytate as the sole source of organic P. Phytase activity correlated positively with the soluble concentration of P (r = 0.72). Phytase activity increased in the presence of NaCl; it was 1.6 and 2.2 times higher at 1.5 and 2.5 M NaCl, respectively, compared to 0 M. Isolates 3 (*Nesterenkonia* sp.), 5 (*Halobacillus* sp.), and 9 (*B. pumilus*) had the highest phytase activity at 0 M of P (90.1, 90.4, and 90.0 µg mL^−1^ h^−1^, respectively). In comparison, isolate 9 (*B. pumilus*) had 120.9 at 1.5 M, and isolate 7 (*B. velezensis*) had 164.7 µg mL^−1^ h^−1^ at 2.5 M NaCl. As observed in alkaline and acid phosphatase activity, isolate 8 (*Halomonas* sp.) had no phytase activity in 0 M NaCl.

The hydrolytic enzyme profile analysis revealed the potential for xylanase production, both in the presence and absence of NaCl, in all bacterial isolates except for *Oceanobacillus* sp. isolates 1 and 2, and for isolate 10 (*Pseudomonas* sp.) (Table 5). Isolate 8 (*Halomonas* sp.) showed no xylanase activity in 0 M NaCl. The presence of NaCl increased xylanase activity in all the producing bacteria, especially at 2.5 M NaCl compared to 1.5 and 0 M. Xylanase activity in 0 M NaCl was not different among the bacteria; however, isolate 11 (*Oceanobacillus* sp.) presented the highest xylanase activity in 1.5 and 2.5 M (1.89 and 5.32 µM mL^−1^ h^−1^, respectively), in contrast to the rest of the isolates. 

All halophilic endophytic bacteria, with the exception of isolates 10 (*Pseudomonas* sp.) and 11 (*Oceanobacillus* sp.), exhibited chitinase activity. Isolate 10, in particular, showed no xylanase or chitinase activity. Similarly, isolate 8 (*Halomonas* sp.) in 0 M NaCl exhibited no activity of both enzymes; however, these two enzymes were active in the presence of NaCl. As with the xylanases, the presence of NaCl enhanced chitinase activity. Chitinase activity in 1.5 and 2.5 M NaCl was consistent among bacteria and varied in the absence of NaCl. These findings demonstrate the highly tolerant nature of chitinase to salt. 

### 2.6. In Vitro Compatibility between Bacterial Isolates

Appendix A shows that there was compatibility between most of the bacteria. However, isolate 6 (*Oceanobacillus* sp.) was incompatible with isolates 5 (*Halobacillus* sp.) and 10 (*Pseudomonas* sp.). Isolate 3 (*Nesterenkonia* sp.) was incompatible with 4 and 7 (*B. velezensis*) and 10 (*Pseudomonas* sp.). 

### 2.7. Principal Component Analysis

Based on the PCA eigenvalues, only 21 variables (of the 42 analyzed) were relevant for selecting halophilic endophytic bacteria in this study (Appendix A). The first three components of the PCA explained 67.4% of the total accumulated variance of the 21 analyzed variables (Figure 5). PC1 accumulated 31.3% of the variance and included the following bacterial traits as the ones with the most significant influence: nitrogenase and phytase enzyme activity, IAA production, P mineralization, Zn and K solubilization, siderophores production, and fructose production (especially in 2.5 M NaCl). PC2 explained 19.6% of the variance and was influenced by IAA production in the absence of NaCl and inorganic P solubilization, acid phosphatase, and citric acid production (both in the absence and presence of salt). PC3 accumulated 16.4% of the variance, and the related variables were nitrogenase enzyme activity in the absence of salt and EPS in both salt conditions. The comparative analysis showed that plant-growth-promoting properties, such as P, K solubilization, phytic acid mineralization, and the production of siderophores, citric acid, and EPS, were positively associated with both NaCl conditions; i.e., the presence of NaCl did not negatively affect these biochemical properties. Endophytic isolates 2 (*Bacillus* sp.), 4 (*B. velezensis*), 5 (*Halobacillus* sp.), 8 (*Halomonas* sp.), and 9 (*B. pumilus*) were of most significant interest due to their favorable response to 2.5 M NaCl. These strains, which are compatible with each other (Appendix A), could be inoculated in glycophyte cultures to mitigate salt stress.

## 3. Discussion of Results

### 3.1. Isolation and Molecular Identification of Endophytic Bacteria by Halotolerance

From the six halophytes under study, 120 endophytic bacteria were isolated. A total of 48.3% were tolerant to 0.1 M NaCl, 23.3% to 0.5 M, 18.3% to 1.5 M and only 9.1% to 2.5 M. Mukhtar et al. [8] also found a lower percentage of isolates tolerant to NaCl as the concentration of NaCl increased in the isolation medium. These authors isolated 49 and 45 bacteria from the halophytes *Atriplex aminicola* and *Salsola stocksii*, respectively. From those, 53% and 52% were tolerant to 0.5 M NaCl, 34% and 40% to 1.5 M, and 13 and 8% to 3 M. From the present research, the latter 11 isolates were considered halophilic based on the classification of salinity-tolerant microorganisms [25].

Genomic analysis showed that these 11 halophilic isolates corresponded to six genera: *Pseudomonas*, *Halomonas*, *Halobacillus*, *Oceanobacillus*, *Bacillus* and *Nesterenkonia* (Figure 1). Only two bacterial isolates, 4 and 7 *(B. velezensis*), belonged to similar species and were isolated from the halophytes *S. torreyana* and *K. scoparia*, respectively. *B. velezensis* was previously reported as an endophyte of the non-halophyte *Lilium leucanthum* [26], while *B*. *pumilus* was reported as an endophyte of the halophyte *Salicornia brachiata* [27]. Mukhtar et al. [8] identified the rhizobacterium *Halobacillus* isolated from the halophyte *Atriplex* sp. The genus *Nesterenkonia* sp. was reported as a halotolerant endophyte isolated from the halophyte tree *Populus euphratica*. Moreover, species of the genus *Halomonas* have been reported as halophilic endophytes from the halophyte *Arthrocnemum macrostachyum* [28]. In addition, Szymanska et al. [29] described *Pseudomonas stutzeri* as a halophyle endophyte isolated from the halophyte *Salicornia europea.*

### 3.2. Indolacetic Acid and Siderophore Production

Production of IAA was observed in the 11 isolates in both NaCl conditions. However, in 9 of the 11 isolates, the IAA production was increased by the presence of 2.5 M NaCl. The highest IAA concentration, in the absence of NaCl, was 12.8 mg L^−1^, and with 2.5 M NaCl, was 20.5 mg L^−1^ (Figure 2a). These concentrations of IAA are lower than those observed by Mahgoub et al. [30], who evaluated the production of IAA in 18 endophytic bacteria isolated from halophytes. These authors observed that all bacteria produced more IAA (between 20 to 70 mg L^−1^) when grown in a nutrient medium with 0.6 M NaCI than without NaCI (8 to 45 mg L^−1^). Other authors [31,32,33] observed similar concentrations to those quantified in the present research in halotolerant endophytic bacteria (*Marinilactibacillus kalidii*, *Stenotrophomonas pavanii*, and *B. licheniformis*) isolated from halophytes, which produced 2.5 mg L^−1^, 20.5 mg L^−1^, and 27.0 mg L^−1^ of IAA, respectively, in the absence of NaCl. However, none of the aforementioned authors quantified IAA bacterial production in the presence of NaCl. 

The isolation of halophilic endophytic bacteria is relevant because they are more adapted to salinity, and the presence of NaCl does not inhibit their capacity to produce auxins. IAA is a significant growth regulator for salt-stressed plants because it controls several growth processes, such as stem elongation, cell division, response to light and gravity, and cell differentiation [34]. Future research should corroborate the effects of IAA production by these halophilic isolates in different plants.

Previous research has qualitatively shown that halotolerant endophytic bacteria produce siderophores [33,35,36], but few studies have tested this ability under saline conditions. In the present research, unlike to IAA, at 2.5 M NaCl, siderophore production increased in five of the 11 bacterial isolates, but no change was observed in the rest of the isolates. The highest siderophore percentage (up to 73%) in both salinity conditions was observed by *Halomonas* sp. (isolate 8) (Figure 2b). Such siderophore production was higher than that reported by Panwar et al. [37] for the halotolerant rhizobacterium *Enterococcus faecium*, which produced 60% in the absence of NaCl and 55% with 0.15 M NaCl; conversely, in *Pantoea dispersa*, the production was 32% and 39%, respectively. Endophytic bacteria produce siderophores, which improve Fe supply and enhance plant growth in saline soils with low Fe availability [38,39,40]. Therefore, endophytic bacteria could help solve this severe agricultural problem; however, future research should confirm the biological relevance of siderophore bacterial production under different environmental conditions.

### 3.3. Nitrogenase Enzyme Activity and Exopolysaccharide Production

In the absence of NaCl, 5 of the 11 endophytic bacteria (45%) had positive nitrogenase enzyme activity. At 2.5 M NaCl, the same bacteria and one more (54%), presented this biochemical property (Appendix A). Shurigin et al. [41] found 5 of 20 bacteria isolated from the halophyte *Haloxylon aphyllum* fixed N_2_ in a semisolid medium without NaCl. Mahgoub et al. [30] observed that all 11 bacteria isolated from the halophyte *A. macrostachyum* fixed N_2_ in the absence of NaCl and in a saline medium (0.6 M NaCl). Enquahone et al. [42] observed that 81% of the halotolerant endophytic bacterial isolates from *Sporobolus specatus* (21 isolates in total) were positive for N_2_ fixation. Using endophytic bacterial isolates 3, 4, 8, 7, 9, and 10 with diazotrophic capacity could be a promising strategy to improve the growth of salt-sensitive plants. Preliminary evidence shows that bacteria with this trait increase N content in seedlings (see later in Section 3.6). 

The present research explored, for the first time, the production of EPS in halophilic endophytic bacteria isolated from halophytes (Appendix A). In recent years, halotolerant bacteria, particularly rhizobacteria, have been shown to improve plant salinity stress through EPS production [39]. EPS produced by halotolerant rhizobacteria can chelate free Na from the soil, restrict Na entry into plants, support biofilm formation, and improve soil stability. EPS production and biofilm formation are essential characteristics of endophytic bacteria under salinity stress conditions. Due to the ability to bind cations, bacterial EPS restricts the Na available for plant uptake and protects the root from high salt concentrations [43]. In the present research, 36% of the bacterial isolates produced EPS independent of the NaCl condition. Future research should address the production and functional characterization of EPS from endophytes, emphasizing the role of EPS in endophyte–plant interactions under saline conditions in halophyte and salinity-sensitive plants. It is of special interest to deepen the study of *B. velezensis* (isolate 4) because this isolate produced both EPS and nitrogenase activity. Bacterial strains with several biochemical properties are of agricultural interest, especially under abiotic stress [44].

### 3.4. Nutrient Solubilization, Bacterial Growth, and pH Modification 

#### 3.4.1. Inorganic and Organic P

All halophilic endophytic bacteria solubilized P from Ca_3_(PO_4_)_2_; however, the presence of NaCl negatively influenced this ability. The maximum P solubilization without NaCl was nearly 16 mg L^−1^ and 11 mg L^−1^ with 2.5 M NaCl (Figure 3a). Shahid et al. [45] reported that the non-halotolerant endophytic bacterium *Priestia aryabhattai* isolated from wheat enhanced P solubilization with increasing NaCl concentrations. The reported solubilization was 20 mg L^−1^ in the presence of 0.8 M NaCl and 40 mg L^−1^ in 2.5 M NaCl. Moreover, *Serratia rubidaea*, an endophyte isolated from the halophyte *Chenopodium quinoa*, solubilized 350.63 mg L^−1^ of P in the presence of 1.3 M NaCl [35]. Although the reported P concentration is higher than that in the present study, the analysis period was ten days instead of the 48 h tested in the present study. Mahgoub et al. [30] observed that the endophytic bacterial strains BR1 (*B. subtilis*) and AR5 (*B. thuringiensis*), isolated from the halophytes *A. macrostachyum* and *Spergularia marina*, produced 211.6 mg L^−1^ and 182.5 mg L^−1^ of P, respectively in the presence of 0.6 M NaCl; however, the NaCl concentration used was four times lower than that of the present study. 

In the solubilization P test from Ca_3_(PO_4_)_2_, all bacteria decreased the pH of the culture medium in both NaCl conditions (Figure 3b). Mahdi et al. [35] also observed a decrease in pH (to 3.0 and 5.9) without NaCl and with 1.3 M NaCl. In the present study, no drastic pH changes were observed in the tests without salt, as observed by these authors. In all bacteria, the pH changes were from 6.8 to 5.9; only *Pseudomonas* sp. (isolate 10) reduced the pH (average 4.5) in both NaCl conditions. 

In the solubilization of P, the production of organic acid varied according to bacteria or NaCl conditions. Citric and succinic acids were commonly secreted by all bacteria in both NaCl conditions. In contrast, the production of lactic acid was observed in all bacteria without NaCl, but three bacteria were not able to produce it with 2.5 M NaCl. Meanwhile, tartaric acid was produced only by three bacterial isolates in both NaCl conditions (Table 1). Paredes-Mendoza and Espinosa-Victoria [46] indicated that the predominant organic acids reported in the solubilization of inorganic P are succinic, citric, lactic, and tartaric acids. In this research, fumaric, oxalic, vanillic, malic, salicylic, and maleic acids were not detected in the extracts produced by the bacteria in the solubilization of nutrients.

Like the solubilization of P from Ca_3_(PO_4_)_2_, all halophilic endophytic bacteria, except isolate 4 (*B. velezensis*), were able to solubilize phytate. Similarly, the presence of NaCl negatively influenced P solubilization, except in *Oceanobacillus* sp. (isolate 1) and *B. velezensis* (isolate 4) (Figure 3c). The outstanding fact was that much higher P solubilization was observed with this organic P source than with the inorganic insoluble P source. The results are also relevant, as most research has focused on evaluating solubilization and mineralization in halotolerant rhizobacteria, but limited information has been generated with halophilic endophytic bacteria. In this context, Liu et al. [47] reported that three halotolerant rhizobacteria (*Pseudarthrobacter*, *Acinetobacter*, and *Pseudomonas*) solubilized more inorganic P (Ca_3_(PO_4_)_2_) than organic P (phytic acid) under salt-free conditions. In the present research, the maximum P-soluble concentrations in the absence of NaCl were, on average, 75 mg L^−1^ with bacteria 3 (*Nesterenkonia* sp.) and 5 (*Halobacillus* sp.), while 2.5 M of NaCl was, on average, 65 mg L^−1^ with isolates 3 and 9 (*Bacillus pumilus*). 

Other investigations highlight the mineralization of organic P in non-halotolerant rhizobacteria. For example, Rasul et al. [48] reported that *Pantoea* sp. solubilized (20 mg L^−1^) and mineralized P (100 mg L^−1^), while *Ochrobactrum* sp. was able to solubilize P (60 mg L^−1^) and mineralize P (150 mg L^−1^).

Insoluble organic P accounts for approximately 65% of total P in saline soils, while inorganic P accounts for 35%. Organic P is present in different forms, including inositol phosphate, phosphomonoesters, phosphodiesters (phospholipids and nucleic acids), and phosphotriesters [49]. Therefore, the P mineralization process exhibited by some microorganisms, such as bacteria, is essential in saline soils. Bacteria mobilize insoluble organic P through mineralization and convert it into available P in the soil [50]. 

In the present research, no significant modification in the pH of LB broth was observed due to phytate solubilization (Figure 3d). P mineralization occurs by mechanisms other than the production of organic acids, which modify the pH of the culture medium [51]. For example, phytases, a subset of phosphatases, gradually dephosphorylate phytate to produce inositol and soluble P [52]. Alkaline and acid phosphatases also mineralize P [49]. The results of the present study suggest that halophilic endophytic bacteria may have a high potential for application in cropping systems with organic fertilization and salinity problems. Hence, future research should test their influence on these conditions. 

#### 3.4.2. Solubilization of K, Mn, and Zn

Some studies have qualitatively evaluated K solubilization with endophytic bacteria isolated from halophytes [53,54]. However, the present research is the first one quantifying the K solubilization capacity of halophilic endophytic bacteria isolated from halophyte roots. In the present research, bacterial K solubilization was positively influenced by NaCl in 6 of 11 isolates. The highest K solubilization with 2.5 M NaCl was observed in isolate 4 (*B. velezensis*), and at 0 M NaCl with isolate 7 (*B. velezensis*) with approximately 57 and 15 mg L^−1^, respectively (Figure 4a). These values are lower in relation to halotolerant rhizobacterium. Ranawat et al. [55] observed that *E. hormaechei* solubilized 98 mg de K L^−1^ in the absence of NaCl. However, in the presence of NaCl, the concentrations of soluble K obtained in the present research are higher than those in other investigations with halotolerant rhizobacteria considering the presence of NaCl. Ashfaq et al. [56] found that K solubilization in the presence of NaCl decreased in the 13 halotolerant rhizobacteria studied; the maximum K solubilization was 22 mg L^−1^ in the absence of NaCl, and 17 with 0.7 M NaCl.

The bacterial ability to solubilize K is relevant because mineral soils generally contain between 0.04% and 3% of K in the first 20 cm of soil. Ninety-eight percent of the total K is mineral, unavailable to plants and microorganisms. Soil microorganisms use mechanisms to solubilize minerals (illite, micas, and feldspars) containing K, which plants can absorb [57]. Studies that use K-solubilizing bacteria to inoculate salinity-sensitive crops are important because this element can compete with high Na concentrations in the soil. Hence, tests under different environmental conditions should be performed to confirm the K solubilization of the halophilic bacteria isolated in this study.

Our understanding of the mechanisms employed by halophilic endophytic bacteria to solubilize K remains limited. However, the most well-known mechanisms include pH decrease through the production of organic acids and proton release. In the present research, all bacterial isolates decreased pH during K solubilization. It is recognized that the organic acids produced to dissolve potassium minerals (illite and feldspar) are oxalic, gluconic, tartaric, 2-ketogluconic, citric, malic, succinic propionic, lactic, acetic, glycolic, malonic, and fumaric acids [58]. In the present research, tartaric, malic and citric acids were common in the solubilization of K in all bacteria; but isolate 8 (*Pseudomonas* sp.) also secreted vanillic acid and was the only one that produced it (Table 2). Vanillic acid production was previously reported in other *Pseudomonas* sp. strains but had not been related to the solubilization of K. Moreover, vanillic acid has been associated with resistance against phytopathogens [59]. Therefore, it is necessary to test whether isolate 8 (*Pseudomonas* sp.) can be used in biocontrol.

In the present study, solubilization of Mn was observed in all the isolates (Figure 4c). Yamaji et al. [60] and Dixit et al. [61] qualitatively analyzed the ability of non-halotolerant endophytic bacteria to solubilize Mn. Microorganisms reduce Mn^4+^ to Mn^2+^ through the production of protons and electron-transporting reducing agents that are oxidized [62], as shown below in Equation (1). The following equation represents the Mn^2+^ release reaction, in which there is no change in the oxidation state: (1)$$MnO_{2}+{4H}^{+}+{2e}^{-}\underset{aq}{\to }{Mn}^{2+}+H_{2}O$$

The bacterial trait to solubilize Mn is relevant due to the influence of soil salinity and sodicity on Mn biogeochemistry [63,64]. Manganese is an essential plant micronutrient, but its availability is influenced by soil pH, redox conditions and salinity. A high concentration of NaCl precipitates Mn, and, consequently, reduces its availability [65]. 

Similar to P and K, no consistent NaCl effect was showed on bacterial Mn solubilization. The presence of NaCl negatively influenced the solubilization in 4 of the 11 bacterial isolates. The highest soluble concentration was observed by the *Hallobacillus* sp. in both NaCl conditions, which was on average 1.33 mg mg L^−1^. This concentration was low and only comparable to research followed by Ijaz et al. [66]. These authors reported that the non-halotolerant rhizobacterium *Bacillus* sp. isolated from maize solubilized 10.7 mg Mn L^−1^ from MnO_2_ at 48 h. 

pH reduction was observed by Mn solubilization for all bacteria. These results agree with other reports on Mn solubilization by pH decrease [62]. However, another solubilization mechanism may occur by utilizing MnO_2_ instead of oxygen as a final electron acceptor in the bacterial respiratory chain [67]. This mechanism remains to be tested in the bacteria evaluated in this study.

Organic acid production is also related to Mn solubilization because this causes pH reduction and the conversion of MnO_2_ to Mn^2+^, which is available to plants [62]. Ijaz et al. [66] reported that the following organic acids were involved in Mn solubilization: formic, oxalic, salicylic, pyruvic, citric, and malic acids. In contrast, in the present research, only two organic acids (citric and malic) were detected in the Mn solubilization (Table 2). This is also different to P, K and Zn solubilization where diverse organic acids are secreted. Similar to the solubilization of inorganic P and K, all bacteria secreted citric acid in the absence and presence of NaCl. Isolate 10 (*Pseudomonas* sp.) produced the highest concentration of citric acid with and without NaCl (15.8 and 16.0 mg L^−1^, respectively). The highest concentration of malic acid was secreted by the same isolate at 0 M NaCl, but at 2.5 M NaCl, three isolates (9, 1 and 2) had the highest Mn soluble concentrations (between 6.8 to 7.4 mg L^−1^). 

Soluble Zn was observed in the 11 isolates. NaCl presence influenced ZnO solubilization in 8 of the 11 isolates. For example, isolate 4 (*B. velezensis*) had the highest solubilization in the absence of NaCl (107 mg L^−1^); however, it significantly decreased with 2.5 M (2.5 mg L^−1^) (Figure 4e). These values are higher than those observed by Fatima et al. [68] in halotolerant rhizobacteria *Alcaligenes* AF7; this bacterium solubilized Zn to 2.79, 3.26, and 2.8 mg L^−1^ with 0, 0.3, and 0.7 M NaCl, respectively. According to Tewari and Arora [69], saline stress may be an important factor for nutrient solubilization in some halotolerant bacterial isolates. The results observed in the present investigation on Zn solubilization are relevant for agriculture because the availability of this element is fundamental for plant nutrition and the activity of several enzymes. As with other micronutrients, soil salinity also negatively influences Zn plant absorption [70]. 

Organic acid production may be one of the mechanisms involved in Zn solubilization [71] that is associated with a decreased pH. This is in accordance with the results obtained in the present research. With all isolates, pH decreased in the Zn solubilization tests, independently of NaCl influence. Upadhyay et al. [72] noted that the main acids involved in Zn solubilization are oxalic, ferulic, caffeic, gallic, syringic, citric, 2-ketogluconic, gluconic, tartaric, maleic, and fumaric acids. These authors detected oxalic, maleic, tartaric, and fumaric acids in ZnO solubilization by non-halotolerant rhizobacteria. Li et al. [73] also identified oxalic, formic, tartaric, and acetic acids in the solubilization of ZnO by rhizobacteria, but information is scarce for halotolerant endophytes. For the first time, in the present study, it was demonstrated that malic acid also participates in Zn solubilization. In both conditions of NaCl, all bacteria produced maleic and malic acids; only three bacteria produced citric acid and six bacteria secreted tartaric acid. The concentration of citric acid and tartaric acid was not influenced by NaCl conditions. The highest citric acid concentration was 9.3 mg L^−1^ in the isolate 8 (*Halomonas* sp.) (Table 2). This value is higher than that observed by Mumtaz et al. [74] in *Bacillus* sp. (3.5 mg L^−1^) and by Zaheer et al. [75] in *Pseudomonas* sp. (4. mg L^−1^), both in the absence of NaCl. In contrast, the NaCl effect on malic and maleic acids concentration was bacteria dependent. The highest malic concentration (4.3 mg L^−1^) at 0 M NaCl was observed in isolate 5 (*Halobacilus* sp.) while at 2.5 M NaCl (4.7 mg L^−1^) was also in isolate 5, and in isolate 3 (*Nesterenkonia* sp.). The highest maleic concentration (11.8 mg L^−1^) at 0 M NaCl was produced by isolate 4 (*B. velezensis*) and at 2.5 M by isolate 4 and *Pseudomonas* sp. (isolate 9) with 10.4 mg L^−1^. To our knowledge, there are no available reports regarding the concentrations of organic acids other than citric acid produced by bacteria in Zn solubilization. Therefore, comparisons are not possible. 

#### 3.4.3. Interaction between Organic Acids and the Solubilization Ion

Research to evaluate the solubilization capacity of bacteria generally focuses on a single ion, while few studies evaluate the production of organic acids involving two or more solubilizing ions. In the present work, the production of organic acids was analyzed for four single solubilizing ions. A Venn diagram shows that, in each solubilization ion, the kind of organic acid was independent of the NaCl conditions; however, as shown in the last section, the concentrations of organic acids were influenced by NaCl (Appendix A). Future research should investigate the type and concentrations of organic acids secreted, as well as the mechanisms involved, in the solubilization of different nutrients at the same time, in order to detect possible interferences or synergies.

#### 3.4.4. Production of Sugars in Nutrient Solubilization Tests

Research on bacteria’s nutrient solubilization capacities has tested the effect of different carbon sources (glucose, fructose, and sucrose); however, no work has evaluated the exudation of sugars in the nutrient solubilization nor the presence or absence of NaCl. The present research, for the first time, shows that sugars may be involved in the solubilization of essential nutrients. Moreover, it indicates that NaCl may influence the concentration of sugars. Although the highest fructose production was observed in the presence of inorganic P, some hypotheses exist regarding the role of fructose in this nutrient´s mineralization. Zhang et al. [76] indicated that fructose is not only a carbon source but also a signaling molecule that triggers the bacterially mediated mineralization processes of organic P. These authors observed that fructose stimulates the expression of phosphatase genes in bacteria and the rate of phosphatase release into the growth medium by regulating their protein secretory system. Fructose was common in 10 of the 11 bacterial isolates in Ca_3_(PO_4_)_2_ solubilization, and twice as much average fructose concentration was observed with 2.5 M NaCl than with 0 M NaCl in the inorganic P solubilization. The highest fructose production was observed in isolates 4 and 7, which were identified as *B. velezensis*. 

In the present research, fructose was also identified in the organic P, K, Mn and Zn solubilization tests. A similar form for inorganic P solubilization, the effect of NaCl on fructose production was isolate-dependent in the solubilization of the rest of the insoluble nutrient forms. The present study is the first to generate the involvement of fructose in the solubilization of diverse essential elements relevant to plant growth and the influence of salinity conditions. Future research should focus on a better understanding of the mechanisms involved.

In contrast to fructose, xylose was less involved in the bacterial solubilization of inorganic P, Mn and Zn. Sharma et al. [77] reported the bacterial production of xylose and its subsequent conversion to aldonic acid, following the transformation of glucose by the enzyme glucose dehydrogenase. This acid can participate in the efficient solubilization of inorganic P. Contrary to expectations, in the inorganic P solubilization, xylose was produced only by isolate 5 (*Halobacillus* sp.) and 10 (*Pseudomonas* sp.) at 0 M NaCl. However, the xylose concentrations produced were very different between these two isolates (29 and 229 µg µL^−1^, respectively). At 2.5 M NaCl, isolate 5 produced (228 µg µL^−1^) and isolate 8 (*Halomonas* sp.) produced only 4.4 µg µL^−1^. Thus, the present investigation suggests the participation of xylose not only in P solubilization, but especially in that of K solubilization (Table 3). Nine of the 11 isolates produced xylose in K solubilization in both NaCl conditions, and 4 of the 9 isolates had comparable and consistent xylose concentrations (between 65 to 73 µg µL^−1^). Less bacteria produced xylose in the solubilization of phytate, Mn and Zn. Future studies should further analyze the participation of xylose in the solubilization of other nutrients and the mechanisms related.

### 3.5. Enzyme Activity in Different Salinity Conditions

Alkaline and acid phosphatase enzyme activity was produced by all the halophilic endophytic bacteria under the three NaCl conditions tested (0, 1.5 and 2.5 M), except in *Halomonas* sp. (isolate 8) at 0 M NaCl. Similarly, *Oceanobacillus* sp. (isolate 1), *Bacillus* sp. (isolate 2) and *Nesterenkonia* sp. (isolate 3) did not produce acid phosphatase at 0 M NaCl. The extracellular activity of alkaline and acid phosphatase enzymes in halotolerant rhizobacteria has been reported [78]. Shabaan et al. [79] also reported on the extracellular enzyme activity of alkaline and acid phosphatase in 25 isolates of halotolerant rhizobacteria. In the present research, two contradictory responses were observed due to the influence of NaCl. While NaCl significantly decreased the alkaline phosphatase activity, the activity of acid phosphatase increased. In contrast, Bylund et al. [80] observed that the enzymatic activity of both phosphatases in *Halomonas elongata* increased twice at 1.3 M NaCl in relation to 0.05 M.

The highest alkaline phosphatase activity (34.45 µg mL^−1^ h^−1^) at 0 M NaCl was produced by *Bacillus* sp. (isolate 2). This activity is lower than that reported by Shabaan et al. [79] in the halotolerant rhizobacterium SMH-7 in 0 M NaCl (164 of µg mL^−1^ h^−1^). In the present research, at 2.5 M NaCl, *Pseudomonas* sp. (isolate 10), *Halomonas* sp. (isolate 8), *B. pumilus* (isolate 9) and *Nesterenkonia* sp. (isolate 3) had, similarly, the highest alkaline phosphatase activity (between 9.55 to 7.75 µg mL^−1^ h^−1^) (Table 4). The activity was higher than that observed by Barrera et al. [81] in the non-halotolerant rhizobacterium *Kosakonia radicincitans* in 0.1 and 2.2 M NaCl (1 µg mL^−1^ h^−1^ at both salt concentrations).

In relation to acid phosphatase activity, the highest concentration (51.32 µg mL^−1^ h^−1^) at 0 M NaCl was observed with *Pseudomonas* sp. (isolate 10) and at 2.5 M NaCl with both *Halobacillus* sp. (isolate 5) and *B. velezensis* (isolate 4), with a concentration of 62.43 and 57.82 µg mL^−1^ h^−1^, respectively. These concentrations were lower than those reported by Shabaan et al. [79] in a halotolerant rhizobacteria (356.93 µg mL^−1^ h^−1^) but higher (2 µg mL^−1^ h^−1^) in two NaCl concentrations (0.1 y 2.2 M) than those produced by the halotolerant bacteria *K. radicincitans* [81]. 

Alkaline phosphatase activity had a low correlation (r = 0.32) with inorganic P, while acid phosphatase had a medium correlation (r = 0.62), which suggests that, for P solubilization, halophilic endophytic bacteria release acid phosphatase in addition to producing organic acids. At pH < 7, the acid phosphatase enzyme shows higher activity [82]. Moreover, a positive correlation between fructose production and acid phosphatase activity was observed in 0 (r = 0.62) and 2.5 M NaCl (r = 0.64); however, this was not the case for alkaline phosphatase. This information is relevant because it shows that inorganic P solubilization by halophilic endophytic bacteria may occur by different mechanisms, as follows: pH decrement, organic acids, phosphatases (mainly acid ones) and fructose participation. 

In the present study, phytase was analyzed because it transforms P from phytic acid to soluble P [52]. All bacteria produced phytase at the three NaCl conditions, except *Halomonas* sp. (isolate 8) at 0 M. In this same NaCl condition, *B. velezensis* (isolate 4) produced the lowest phytase concentration (4.24 µg mL^−1^ h^−1^). Remarkably, these two isolates activated phytase production in the presence of 2.5 M NaCl with 117.74 and 144.94 µg mL^−1^ h^−1^ for isolates 4 and 8, respectively. The highest average phytase concentrations at 0 M NaCl were observed in three isolates: *Nesterenkonia* sp. (isolate 3), *Halobacillus* sp. (isolate 5) and *B. pumilus* (isolate 9) (Table 4). Phytase activity in these bacteria was higher than that observed in the halotolerant rhizobacterium *P. azotoformas* (10.08 µg P mL^−1^ h^−1^) in 0 M NaCl [83]. Remarkably, in the present research, *B. pumilus* (isolate 9) at 2.5 M NaCl produced the highest phytase concentration (154.42 µg P mL^−1^ h^−1^). 

Halophilic endophytic microorganisms associated with halophytes are a potential source of several hydrolytic enzymes that are functional in saline environments such as xylanases and chitinases. Xylanase activity was quantified in *Nesterenkonia* sp., *B. velezensis* (the two isolates), *Halobacillus* sp., *Oceanobacillus* sp., *B. pumilus* and *Pseudomonas* sp. (Table 5). Xylanase activity in the genus *Oceanobacillus* was qualitatively observed by Rohban et al. [84]. In the present study, three (isolates 1, 2 and 10) from 11 bacterial isolates did not produce xylanase in any NaCl conditions analyzed in the present research (0, 1.5 and 2.5 M). Wejse et al. [85] referred to these enzymes as multi-extremophiles because they are functional in high salt concentrations (>1 M NaCl), which is not the case for most other proteins. This is in accordance with the xylanase production behavior in most of the isolates of this study; their activity increased as NaCl concentrations were enhanced. *Halomonas* sp. (isolate 8) only secreted xylanases in the presence of NaCl (Table 5).

Khan et al. [86] showed that the halotolerant endophyte bacterium *Bacillus* sp. TKE4 showed xylanase activity in the absence of NaCl (9.4 µM mL^−1^ h^−1^), which is higher than the average activity found in the present research in *B. velezensis* (isolate 4) with the highest xylanase activity at 0 NaCl (0.14 µM mL^−1^ h^−1^). Giridhar et al. [87] reported that the halotolerant endophyte *Gracilibacillus* sp. displayed xylanase activity in 0 and 2 M NaCl (0.9 and 2.8 µM mL^−1^ h^−1^, respectively).

In the present study, at 2.5 M NaCl, the isolate 11 (*Oceanobacillus* sp.) had higher xylanase activity (5.32 µM mL^−1^ h^−1^). The halotolerant bacterium *Marinimicrobium* sp. maintained the same xylanase activity (60 µM mL^−1^ h^−1^) at two concentrations of NaCl (1.7 and 3.5 M). Xylanase production by endophytic bacteria allows them to degrade plant cell walls and leads to colonization. Xylanases are also related to the biological control of phytopathogenic fungi [88]. Therefore, bacteria with xylanase activity could be used as microbial inoculants for biological control. However, future research should test this potential bacterial trait under in vitro, in vivo and field conditions. 

Production of chitinase was observed in all bacteria at 0 M NaCl except bacteria 8 (*Halomonas* sp.), 10 (*Pseudomonas* sp.) and 11 (*Oceanobacillus* sp.). These two later isolates did not produce chitinase either at 2.5 M NaCl, but isolate 8 and the other nine did (Table 5). Finding enzymes that show activity at various salt concentrations is relevant because salt does not limit their stability [89]. The rhizobacterium *B. subtilis* had chitinase activity of 120 µM mL^−1^ h^−1^ at 120 h in 0 M NaCl [90]. This activity was much higher than that exhibited by the bacteria in the present research analyzed at an incubation time of 48 h (between 1.24 and 1.90 µM mL^−1^ h^−1^). The halotolerant rhizobacterium *Planococcus rifitoensis* had chitinase activity of 10 µM mL^−1^ h^−1^ in 0 M NaCl, while in 1.7 M NaCl, it decreased to 6 µM mL^−1^ h^−1^. This concentration is comparable to those quantified in the present research: from 5.68 to 6.92 µg mL^−1^ h^−1^ at 1.5 M NaCl and from 5.45 to 6.70 µg mL^−1^ h^−1^ at 2.5 M NaCl. These findings indicate that, similar to bacteria with xylanase activity, bacteria with chitinase activity could be used in the biological control of phytopathogenic fungi [90]. As mentioned before, the activity of these enzymes should be tested in further studies, as this was not an objective of the present research. 

### 3.6. In Vitro Compatibilty between Bacterial Isolates

A relevant complementary bacterial trait was that several halophilic isolates (7 from 11) were compatible with each other. In contrast, *Oceanobacillus* sp. (isolate 6) was incompatible with two isolates: *Halobacillus* sp. (isolate 5) and *Pseudomonas* sp. (isolate 10). At the same time, *Nesterenkonia* sp. (isolate 3) was not compatible with the following three isolates: *B. velezensis* (isolates 4 and 7), and *Pseudomonas* sp. (isolate 10) (Appendix A). Bacterial compatibility is critical for formulating inoculants that promote plant growth [91]. A bacterial consortium generally consists of two or more compatible bacteria of different species in a complementary or synergistic interaction. Khan et al. [92] demonstrated that a consortium of *Bacillus* sp., *B. subtilis* and *Bacillus cereus* improved synergistic wheat growth under saline stress more than a single bacterial inoculation. This compatibility analysis was also useful for the selection of the most promising bacterial isolates (see next section), based on these main biochemical traits, to be tested in future experiments in glycophyte plants.

### 3.7. Principal Component Analysis

The principal component analysis showed the most relevant biochemical parameters (21) for the selection of bacterial halophiles derived from halophytes distributed in three main compartments with significant accumulated variance (67.4%) (Figure 5). Additionally, it demonstrated that the plant-growth-promoting properties. such as P, K solubilization, phytic acid mineralization, and the production of siderophores, citric acid, and EPS, were positively associated with both NaCl conditions; i.e., the presence of NaCl did not negatively affect these biochemical properties. Halophilic endophytic isolates 2 (*Bacillus* sp.), 4 (*B. velezensis*), 5 (*Halobacillus* sp.), 8 (*Halomonas* sp.), and 9 (*B. pumilus*) were of significant interest because they had a remarkably better biochemical performance than the rest of the bacteria analyzed in this study. Moreover, these traits had favorable responses to 2.5 M NaCl and were compatible with each other (Appendix A). The promising results from the present research support the elaboration of a bacterial consortium that could potentiate these bacteria´s beneficial functions. Designing bacterial consortia has gained interest as a suitable strategy for sustainable agricultural production [93]. In this context, the endophytic bacteria isolated from halophyte roots showed functional complementarity in plant-growth-promoting properties that may promote phytostimulation, which contributes to host plant productivity under salt stress, as suggested by Gaiero et al. [94]. A running experiment validates the positive effect of this consortium, with bacteria that produced IAA and had positive nitrogenase activity, in tomato seedlings inoculated through seed priming. The validation of other bacterial traits and consortia building still requires further testing. Therefore, crop inoculation with this consortium may be a viable strategy for sustainable crop production in salinity-based agriculture, including crop production in arid and semi-arid environments.

## 4. Materials and Methods

### 4.1. Selection of Halophytes, Sampling, and Root Disinfection 

Fine roots of six dominant halophytes (*Distichlis spicata*, *Cynodon dactylon*, *Eragrostis obtusiflora*, *Suaeda torreyana*, *Kochia scoparia*, and *Baccharis salicifolia*) were sampled in an area in the east of the State of Mexico in the former Lake Texcoco (19.27°N, 98.54°O and 2236 masl; Appendix A). The site is characterized by its salt content [95]. Samples were placed in separate plastic bags inside a cooler (4 °C) and immediately transported to the laboratory for analysis.

Roots were disinfected according to the methodology of Albdaiwi et al. [96]. Briefly, the roots were washed under running water for 10 min to remove adhering soil particles, placed in 70% ethanol for one minute, and rinsed three times with sterile distilled water. A 3% sodium hypochlorite solution was added and left for five min, followed by six rinses with sterile distilled water. After the sixth rinse, an aliquot (100 μL) of wash was seeded onto a Luria Bertani (LB) culture medium and incubated at 28 °C for 15 days. Surface disinfection was considered successful when no growth was observed on the medium.

### 4.2. Isolation of Endophytic Bacteria by Halotolerance

The isolation of bacteria from the roots of each halophyte was performed by halotolerance in LB media, with increasing concentrations of NaCl (0.1, 0.5, 1.5, and 2.5 M). Initially, 0.5 g of the root tissue was macerated in a mortar with 10 mL of sterile distilled water. Subsequently, serial decimal dilutions were prepared, and 100 µL of the last dilution (10^−3^) was placed in Petri dishes prepared with the LB medium at different NaCl concentrations. The Petri dishes were incubated for 15 days at 28 °C [2]. Bacteria isolated at the 2.5 M NaCl concentration were purified, molecularly identified, and biochemically characterized. Bacteria grown at the other NaCl concentrations were cryopreserved in 20% glycerol for future use.

### 4.3. Molecular Identification of Halophilic Endophytic Bacteria

DNA isolation was performed according to Töpper et al. [97]. Bacteria were placed in an LB medium and incubated at 28 °C for 48 h at 160 rpm. Then, 1 mL of each bacterial suspension was placed in an Eppendorf tube and centrifuged at 14,000 rpm for 3 min. Subsequently, DNA extraction was performed with the commercial DNeasy Blood & Tissue Kit from Qiagen (Hilden, Germany), according to the manufacturer’s instructions. The 16S rDNA gene fragment was amplified by polymerase chain reaction (PCR) with the conditions shown in Appendix A. The universal primers were 16SF direct: 5′-GCCTAACACATGCAAGTC-3′ and 16SR reverse: 5′-AAGGAGGTGATCCAGCCGCA-3′. A reaction mixture was prepared for each oligonucleotide pair, as described in Appendix A. The PCR product was approximately 1500 bp, which was verified on 1% agarose gel. Subsequently, the 16S rRNA sequences were compared with the GenBank database using the NCBI BLAST nucleotide search. A multiple sequence alignment was constructed with the ClustalX 1.8 software package (http://www.clustal.org/clustal2, accessed on 11 March 2024), and a phylogenetic tree was prepared using the neighbor-joining method in MEGA v6.1 software (www.megasoftware.net), with confidence tested by bootstrap analysis (1000 replicates).

### 4.4. Biochemical Characterization of Halophilic Endophytic Bacteria

Halophilic bacteria were characterized as follows: according to the production of indole acetic acid (IAA), siderophores, and exopolysaccharides (EPS); nutrient solubilization (P, K, Zn, and Mn); phytic acid mineralization; nitrogenase enzyme activity; production of organic acids and sugars in the solubilization broth of all nutrient solubilization tests; and enzyme activity (acid and alkaline phosphatase, phytases, xylanases, and chitinases).

For all evaluations, bacteria were propagated in the LB medium supplemented with 2.5 M NaCl and incubated for 48 h at 28 °C. Bacterial cultures were adjusted to 0.5 absorbance at 600 nm (equivalent to 1 × 10^6^ bacteria mL^−1^). From each bacterial suspension, one mL was taken and placed in 10 mL of a prepared LB broth to measure the solubilization of each nutrient, siderophore production, and IAA separately. These tests were performed in the presence (2.5 M) and absence of NaCl.

#### 4.4.1. Indole Acetic Acid and Siderophore Production

IAA production was measured using the procedure described by Bano and Musarrat [98] in LB broth supplemented with 1 mg L^−1^ of L-tryptophan. Samples were incubated at 35 °C for 48 h and then centrifuged at 9000 rpm for 10 min. One mL of the supernatant was placed in a test tube, and two mL of Salkowski reagent was added. The samples were incubated for 30 min until a pink color was obtained, which is characteristic of IAA production. The absorbance of the sample was measured in a spectrophotometer at a wavelength of 530 nm [99], while the concentration was determined based on a standard curve with concentrations between 12–60 mg L^−1^ of IAA.

The quantification of siderophores was performed using the methodology of Arora and Verma [100]. Bacterial cultures were grown in LB broth and centrifuged at 10,000 rpm for 10 min. The bacterial supernatant (0.5 mL) was mixed with 0.5 mL CAS reagent, and after 20 min, the absorbance was measured at 630 nm. Siderophore production was calculated using the following formula and reported as a percentage [101]:$$Siderophore\;production=\frac{Ar-As}{Ar}\times 100$$
where

Ar = reference absorbance (CAS solution and union culated culture medium), and As = absorbance of the sample (CAS solution and uninoculated supernatant).

#### 4.4.2. Exopolysaccharide and Nitrogenase Enzyme Production

EPS production was evaluated using the method modified by Chaudhari et al. [102]. The LB medium was prepared with 0.8 g L^−1^ of Congo red dye supplemented with 3% sucrose. Bacterial isolates were seeded in Petri dishes with the prepared medium and incubated at 37 °C for 48 h. According to Arciola et al. [103], bacterial colonies that were very dark to almost black were designated as positive for EPS production. 

The ability to fix atmospheric nitrogen was evaluated according to the methodology of Xie et al. [104] in a semisolid sucrose malate medium with bromothymol blue. After one week, the change in color from green to blue was considered positive for nitrogenase activity.

#### 4.4.3. Quantitative Determination of Nutrients Solubilization and Mechanisms Involved 

The methodology described by Battini et al. [105] was used to determine P solubilization and P mineralization. Solubilization was determined from an inorganic source (Ca_3_(PO_4_)_2_), and mineralization was determined from an organic one (phytic acid: C_6_H_18_O_24_P_6_). In addition, NBRIP and PSM (Phytate Screening Medium) culture media were prepared for inorganic P solubilization and organic P mineralization, respectively. K solubilization was measured with the methodology of Parmar and Sindhu [106], using a mineral medium with 3% potassium feldspar. Feldspars are among the most important minerals in soil, and their chemical formula is KAlSi_3_O_8_. Mn and Zn solubilization was evaluated using 50 mM MnO_2_ and 0.1% Zn, respectively, according to the methodology described by Sanket et al. [67].

After inoculation with the bacteria, all the solubilization tests were incubated at 28 °C for 48 h at 120 rpm. Absorbance was determined at 600 nm in a UV–visible spectrometer as an indirect measure of bacterial growth (Cary 50, Varian, Palo Alto, CA, USA). Then, the samples were filtered with Whatman No. 40 filter (Whatman, Maidstone, UK) paper and centrifuged at 9000 rpm for 10 min. The cell-free supernatants were measured for pH (potentiometrically) and soluble nutrient concentrations (P, K, Mn, and Zn). All tests had three replicates for evaluating each solubilization ion, and a treatment without inoculation (control) was included. 

A 0.5 mL aliquot of the bacterial cell-free supernatant was placed in a test tube, and 1 mL of vanadate–molybdate solution and 3.5 mL of deionized water were added to quantify soluble P. The samples were allowed to stand for 10 min until they developed a yellow color. Then, absorbance was measured with a spectrophotometer (Perkin Elmer 3110, Waltham, MA, USA) at a wavelength of 400 nm [107]. The concentration of soluble P was determined by comparing the absorbance of the sample with a standard curve with concentrations ranging from 1–6 mg L^−1^. Soluble K was measured by flame emission photometry (Jenway PFP7, Hong Kong, China). Soluble Mn and Zn concentrations were quantified with an atomic absorption spectrometer (Perkin Elmer 3110). Certified standards were used to prepare the respective calibration curve standards with deionized water (0.2 Mohms).

##### Production of Organic Acids and Sugars

Bacterial cell-free supernatants from inorganic P, K, Mn, and Zn solubilization were frozen until organic acid analysis [46]. Samples were thawed, sonicated for 20 min (at 40 kHz; Branson 1510, Brookfield, CT, USA), and filtered with a 0.22 µm nylon mesh. The samples were then analyzed by HPLC-IR (LDC Analytical IR Detector Varian ProStar, Palo Alto, CA, USA) with a Phenomenex Rezex column (ROA, Organic acid H^+^ 300 × 7.8 mm). The following 12 acids (Sigma, St. Louis, MO, USA) were used as standards: fumaric, citric, oxalic, lactic, acetic, acetic, malic, tartaric, salicylic, maleic, vanillic, and succinic. 

Sugars were determined according to the methodology of Murkovic and Derler [108] in bacteria-free solubilization extracts. The standards used were D-glucose, D-fructose, D-sucrucose, D-lactose, D-maltose, D-xylose, D-arabinose, and trehalose. Calibration curves were prepared with a concentration of 1 mg mL^−1^ of these standards. The HPLC system configuration was 25 water: 75 acetonitrile. In addition, separation of the compounds was performed with a CarboPac PA1 anion exchange column (250 × 4 mm; Dionex Corp., Sunnyvale, CA, USA) and a CarboPac PA1 guard column (50 × 4 mm; Dionex) with a volume of 1 µL and a constant flow rate of 1 mL min^−1^.

#### 4.4.4. Extracellular Enzyme Activity at Different Salinity Conditions

To measure the extracellular enzyme activity of acid and alkaline phosphatase, phytase, xylanase, and chitinase, each bacterium separately was cultured in LB broth without NaCl and with bacterial density adjusted to 0.5 absorbance at 600 nm. Enzyme activity was evaluated at three NaCl concentrations (0, 1.5, and 2.5 M). All enzyme activity assays were performed in triplicate, and a control was included. 

Phosphatase activity was determined using disodium p-nitrophenylphosphate (PNPP 0.025 M) as a colorimetric substrate [107]. Each bacterial isolate was cultured at 28 °C for 48 h in NBRIP medium at pH = 7 supplemented with Ca_3_(PO_4_)_2_ (5 g L^−1^), and 1 mL of adjusted culture medium was added. Cell-free supernatant samples were obtained by centrifuging 2 mL of culture of each bacterial isolate at 10,000 rpm for 10 min. In a test tube, 0.5 mL of NBRIP supernatant, 0.5 mL of substrate (NBRIP medium), and 2 mL of 0.5 modified universal buffer (MUB) were added. The MUB was adjusted to pH = 6.5 for acid phosphatase and pH = 11 for alkaline phosphatase. The reactions were carried out at 37 ± 1 °C for 60 min and stopped by adding 0.5 mL of 0.5 M CaCl_2_ and 2 mL of 0.5 M NaOH. The samples were filtered using Whatman No. 42 filter paper. The p-nitrophenol (PNP) formed was measured spectrophotometrically at 400 nm. Three independent replicates per treatment were performed. A unit of enzyme activity (U) was expressed as micrograms of PNP released per milliliter per hour.

An NBRIP medium supplemented with phytic acid (2 g L^−1^) was prepared for quantitative analysis of extracellular phytase production, and 1 mL of the bacterial suspension adjusted to 0.5 absorbance was added. After incubation at 28 °C for 48 h, the cultures were centrifuged at 10,000 rpm for 10 min, and the supernatant was used for extracellular phytase estimation. Phytase activity determines the amount of inorganic P released into the medium [109]; therefore, two mixtures were prepared to assess the rate of released P. In Mixture 1, the supernatant (0.2 mL) was mixed with 0.5 g of phytic acid dissolved in 100 mL of sodium acetate buffer (0.2 M, pH = 5.5). After 30 min of incubation at 28 °C, the reaction was stopped by adding 10 mL of 15% trichloroacetic acid. In Mixture 2, 100 mL sulfuric acid (1 M) was mixed with 100 mL ascorbic acid (10%) and 4 mL ammonium molybdate (3:1:0.1) 2.5%. For phytase estimation, 100 µL of Mixture 1 was mixed with 900 µL of Mixture 2 and incubated at 28 °C for 1 h. The absorbance of the final mixtures was measured at 400 nm [107]. A unit of enzyme activity (U) was expressed as micrograms of P released per milliliter per hour. 

For quantitative analysis of extracellular xylanase production, an LB broth was prepared with 1% xylan and 1 mL of bacterial culture medium adjusted to 0.5 absorbance. After incubation at 28 °C for 48 h, the cultures were centrifuged at 10,000 rpm for 10 min to obtain the supernatant, which was used for extracellular xylanase estimation. Xylanase activity was estimated by measuring the yield of reducing sugars with modification of the 3,5-dinitrosalicylic acid (DNS) method [110]. The supernatant (50 µL) was mixed with 50 µL of 0.1 M citrate buffer (pH = 4.8) and 1% xylan and incubated at 55 °C for 60 min. Then, 200 µL of DNS solution was added to the reaction, which was maintained in a boiling water bath for 5 min. The released reducing sugars were estimated using a standard curve of 1 mg mL^−1^ glucose.

Chitinase activity was determined by the method of Ramírez et al. [111]. The LB broth medium was supplemented with colloidal chitin (10%), and the bacterial suspension (1 mL), adjusted to 0.5 absorbance, was incubated at 28 °C for 48 h. The bacterial cultures were centrifuged at 10,000 rpm for 10 min to obtain the supernatant. Then, 10% colloidal chitin and 1 mL of supernatant were mixed and incubated at 30 °C for 60 min. The reaction was stopped by adding 1 mL of 1% NaOH. The product was determined by 3,5-dinitrosalicylic acid (DNS) assay, and the absorbance was measured at 535 nm. Chitinase activity was defined as the enzyme required to produce 1 μM of N- acetylglucosamine per hour per mL of supernatant [112].

### 4.5. Compatibility between Bacterial Isolates

For the possible association of bacteria (bacterial consortia), in vitro compatibility between bacterial isolates was tested in triplicate, as described by Sundaramoorthy et al. [113]. Bacterial isolates were inoculated in Petri dishes with LB medium, two by two, in an extended manner. The isolates were distributed in perpendicular lines. The Petri dishes were incubated at 28 °C for 72 h. Compatible bacterial isolates grew on top of each other, while non-compatible isolates formed inhibition areas between combinations.

### 4.6. Statistical Analysis

The Shapiro–Wilk normality test was performed for all quantitative variables, while homogeneity of variances was corroborated with Bartlett’s test (α = 0.05). All variables presented normal distribution; thus, data transformation was not necessary. One-way ANOVA (*p* ≤ 0.05) and Tukey’s post hoc (*p* ≤ 0.05) tests were performed with R software version 4.0.5. Principal component analysis (PCA) was performed with the R package Factoextra version 4.0.5 [114].

## 5. Conclusions

Of the 120 root endophytic bacteria isolated from halophytes, 11 were halophilic and exhibited promising characteristics to improve plant growth, both in the presence and absence of NaCl. Therefore, these bacteria can be used as microbial inoculants to promote plant growth and biostimulate nutrient solubilization, and also be tested for their contribution to sustainable agriculture under salinity conditions due to their adaptation to these conditions. The isolated bacteria had a higher affinity for organic P mineralization compared to inorganic P. Salt concentration increased nutrient solubilization depending on the bacterial strain and solubilization ion. The nitrogenase activity found in some bacteria in the presence of salt is relevant since, under saline conditions, nitrogen limits plant growth and food production. The results showed a relationship between the production of organic acids and the solubilization of nutrients that have low availability in saline soils such as K, Zn, and Mn. An exclusive production of maleic acid in Zn solubilization and fructose and xylose in the nutrient solubilization was observed. The presence of salt influenced fructose but not xylose secretion in some bacteria. The activity of acid phosphatase, phytase, xylanase, and chitinase enzymes increased in the presence of salt, showing stability at the salt concentrations tested. Halophilic endophytic bacteria displayed xylanase and chitinase activity, and these enzymes should be tested in biological control agents to protect crops from phytopathogenic fungi. Finally, the fact that these bacteria are mostly compatible provides a solid basis for the future formulation of consortia with desirable biochemical characteristics to help mitigate salt stress.

## Figures and Tables

**Figure 1 plants-13-01626-f001:**
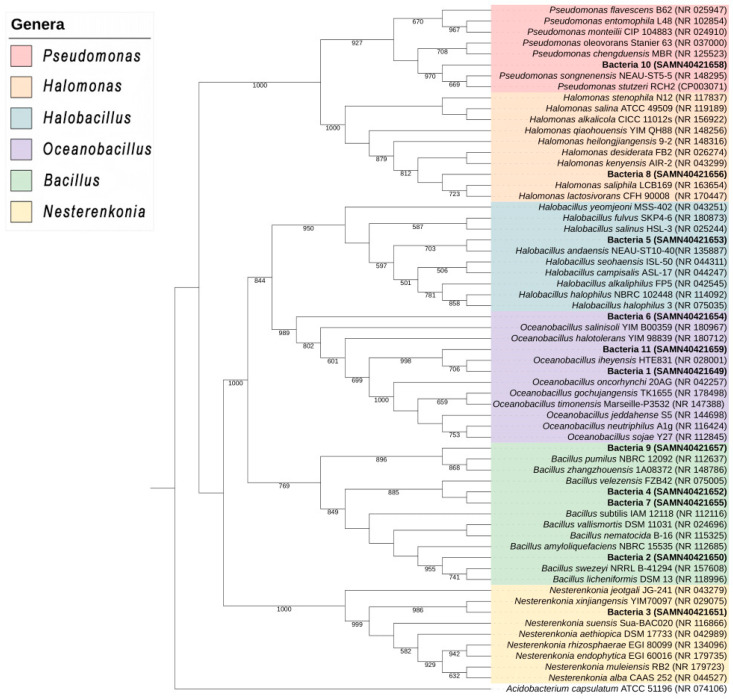
Phylogenetic tree established by the neighbor-joining method based on 16S rRNA sequences and closely related sequences.

**Figure 2 plants-13-01626-f002:**
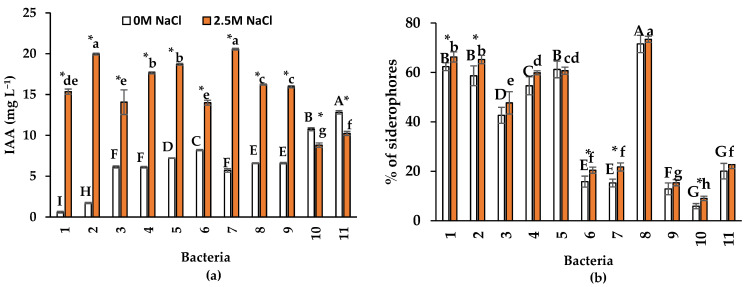
Effect of NaCl on indol acetic acid (**a**); and siderophores (**b**) production by endophytic halophytic bacteria isolated from halophytes. Values correspond to average ± standard deviation, n = 3. Different capital letters show differences in the production of IAA and siderophores when comparing endophytic bacteria in 2.5 M NaCl. Different lowercase letters show differences when comparing endophytic bacteria in 0 M NaCl. Asterisk (*) significantly differs when comparing the same bacteria in 0 or 2.5 M NaCl. In all cases, one-way ANOVA and Tukey’s post hoc test (α = 0.05) were used.

**Figure 3 plants-13-01626-f003:**
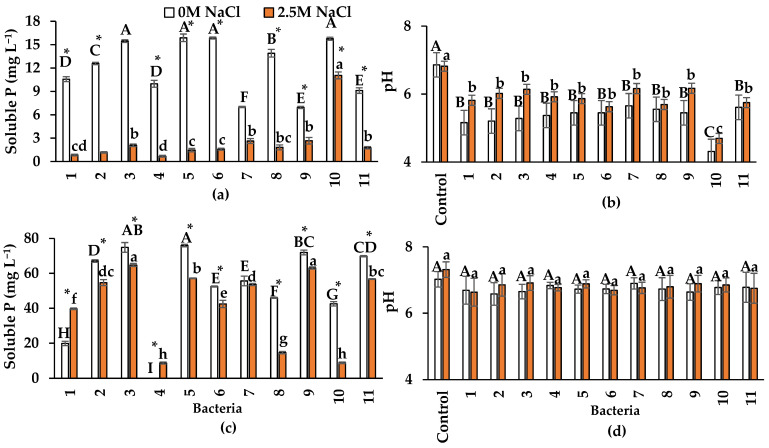
Soluble P concentration and pH variation in the free-bacteria extract after solubilization test (48 h) by halophilic endophytic bacteria in two NaCl concentrations: (**a**,**b**) when using Ca_3_(PO_4_)_2_; or (**c**,**d**) phytic acid. Values correspond to average ± standard deviation, n = 3. Different capital letters show differences in the soluble P concentration and pH when comparing endophytic bacteria in 0 M NaCl. Different lowercase letters show differences when comparing endophytic bacteria in 2.5 M NaCl. Asterisk (*) shows a significant difference when comparing the same bacteria in 0 or 2.5 M NaCl. In all cases, one-way ANOVA and Tukey’s post hoc test (α = 0.05) were used.

**Figure 4 plants-13-01626-f004:**
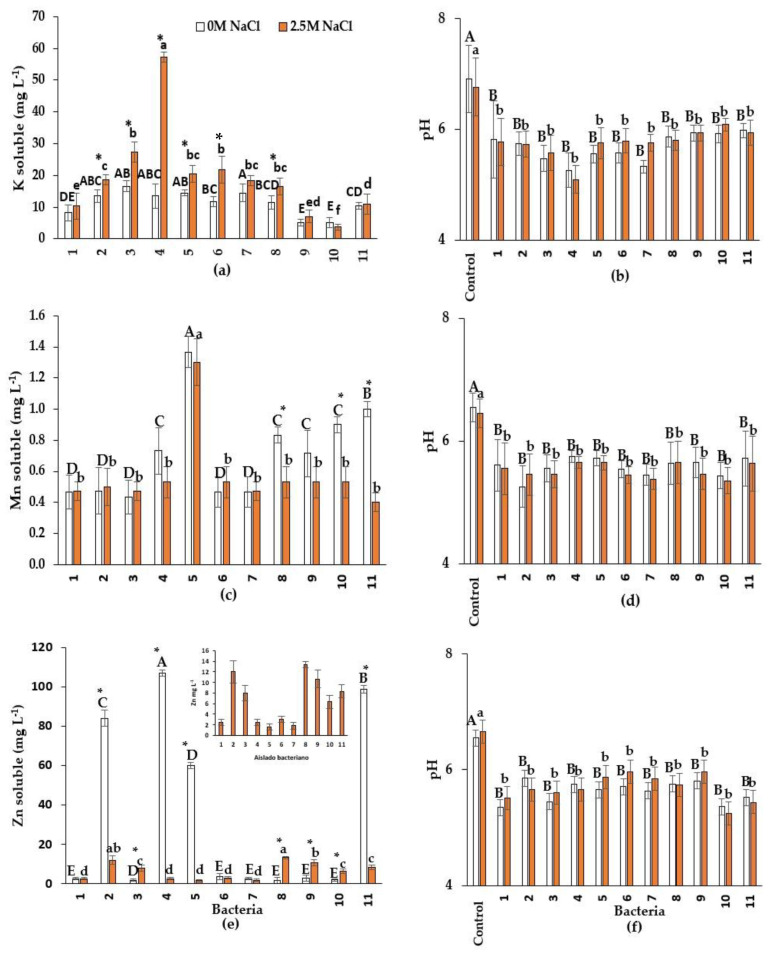
Concentration of soluble of K (**a**,**b**); Mn (**c**,**d**); and Zn (**e**,**f**), and variation in pH in the free-bacteria extract after the solubilization test (48 h) by halophilic endophytic bacteria in two NaCl concentrations. Values correspond to average ± standard deviation, n = 3. Different capital letters show differences in the soluble P concentration and pH when comparing endophytic bacteria in 0 M NaCl. Different lowercase letters show differences when comparing endophytic bacteria in 2.5 M NaCl. Asterisk (*) shows a significant difference when comparing the same bacteria in 0 or 2.5 M NaCl. In all cases one-way ANOVA and Tukey’s post hoc test (α = 0.05) were used; (**e**) a better visualization of results obtained with NaCl is shown.

**Figure 5 plants-13-01626-f005:**
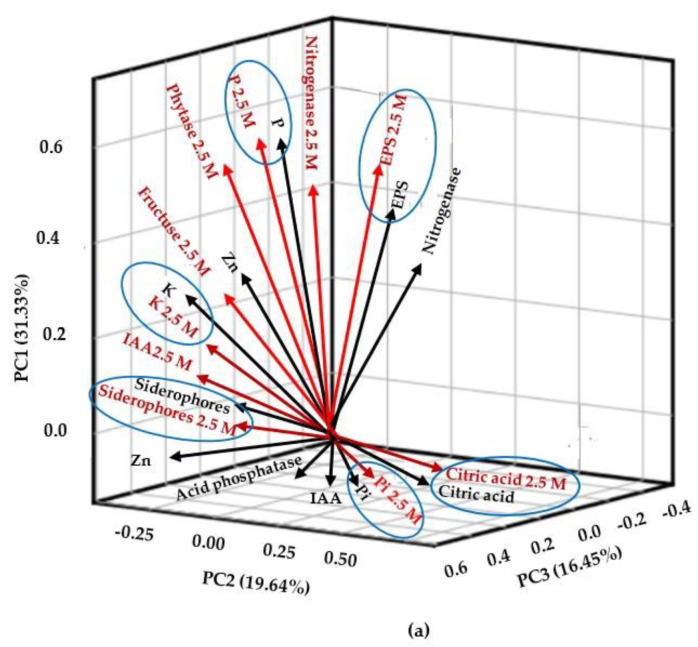
Principal component analysis (PCA) from the biochemical characterization of halophilic endophytic bacteria isolated from six halophytes: (**a**) biochemical variables analyzed in different NaCl concentrations. Blue circles show similar response in both NaCl concentrations tested; and (**b**) distribution of bacteria in the PCA. Variables showed in black for 0 M NaCl and in red for 2.5 M NaCl. Indole acetic acid (IAA), exopolysaccharides (EPS), soluble manganese (Mn), soluble potassium (K), soluble phosphate (Pi) from solubilization with Ca_3_(PO_4_)_2_, soluble phosphate (P) from phytic acid solubilization test, soluble zinc (Zn). The numbers represent the bacterial isolates.

**Table 1 plants-13-01626-t001:** Organic acids (mg L^−1^) produced by halophilic endophytic bacteria in NBRIP-Ca_3_(PO_4_)_2_ broth in the presence or not of NaCl.

	0 M NaCl				2.5 M NaCl			
Bacteria	Citric	Succinic	Lactic	Tartaric	Citric	Succinic	Lactic	Tartaric
1	6.3 ± 0.4 B*	1.8 ± 0.2 A	4.7 ± 0.3 A	nd	7.2 ± 0.1 b*	1.3 ± 0.1 a	4.2 ± 0.2 a	nd
2	6.2 ± 0.4 B*	1.4 ± 0.4 A	5.2 ± 0.3 A*	nd	4.6 ± 0.1 b*	1.0 ± 0.2 a	4.0 ± 0.2 a*	nd
3	1.4 ± 0.2 C	1.2 ± 0.1 A	0.4 ± 0.2 B*	1.8 ± 0.1 A	1.9 ± 0.2 b	1.2 ± 0.1 a	nd *	1.4 ± 0.1 a
4	15.5 ± 0.5 A	0.4 ± 0.1 A	0.8 ± 0.2 B	nd	15.2 ± 0.3 a	0.8 ± 0.3 a	1.2 ± 0.2 b	nd
5	5.6 ± 0.3 B	0.6 ± 0.3 A	1.0 ± 0.1 B	nd	5.2 ± 0.1 b	0.4 ± 0.3 a	1.0 ± 0.2 b	nd
6	5.6 ± 0.6 B	0.8 ± 0.3 A	0.6 ± 0.1 B*	nd	6.5 ± 0.3 b	1.0 ± 0.4 a	1.8 ± 0.2 b*	nd
7	2.2 ± 0.1 C	1.4 ± 0.2 A	0.6 ± 0.2 B*	nd	2.8 ± 0.3 b	1.2 ± 0.4 a	1.4 ± 0.3 b*	nd
8	1.4 ± 0.1 C*	0.4 ± 0.2 A*	1.4 ± 0.2 B*	nd	2.1 ± 0.5 b*	1.0 ± 0.1 a*	nd *	nd
9	2.6 ± 0.1 C	1.0 ± 0.1 A	1.0 ± 0.3 B*	1.4 ± 0.1 A	2.4 ± 0.2 b	1.4 ± 0.1 a	nd *	1.0 ± 0.2 a
10	3.2 ± 0.3 C	1.2 ± 0.1 A	3.2 ± 0.5 A*	nd	3.8 ± 0.2 c	1.0 ± 0.2 a	4.0 ± 0.1 a*	nd
11	3.1 ± 0.1 C	1.0 ± 0.2 A	1.8 ± 0.2 B	nd	3.6 ± 0.1 c	1.0 ± 0.2 a	2.0 ± 0.1 b	nd
Range	1.4–15.5	0.4–1.8	0.4–5.2	0–1.8	1.9–15.2	0.4–1.4	1.0–4.2	0–1.4
Mean	4.8	1.0	1.8	0.2	5.0	1.0	0.4	0.2

Average ± and standard deviation, n = 3. Different capital letters show differences when comparing endophytic bacteria in 0 M NaCl. Different lowercase letters show differences in organic acid production when comparing endophytic bacteria in 2.5 M NaCl. Asterisk (*) shows a significant difference when comparing the same bacteria in 0 or 2.5 M NaCl. In all cases one-way ANOVA and Tukey´s post hoc test (α = 0.05). nd = no detected.

**Table 2 plants-13-01626-t002:** Organic acids (mg L^−1^) produced by endophytic bacteria when solubilizing K, Mn, and Zn that are influenced by 2.5 M NaCl concentration.

	0 M NaCl					2.5 M NaCl				
Bacteria	Citric	Malic	Tartaric	Maleic	Vanillic	Citric	Malic	Tartaric	Maleic	Vanillic
					K					
1	5.2 ± 0.1 B*	0.4 ± 0.1 C	1.7 ± 0.2 B*	Nd	Nd	4.0 ± 0.2 b*	0.8 ± 0.1 b	2.8 ± 0.1 b*	Nd	Nd
2	4.8 ± 0.1 B	0.8 ± 0.1 C*	1.3 ± 0.2 B	Nd	Nd	4.7 ± 0.3 b	1.6 ± 0.2 b*	2.0 ± 0.1 b	Nd	Nd
3	1.9 ± 0.3 C	1.0 ± 0.1 C*	8.9 ± 0.1 A*	Nd	Nd	2.2 ± 0.1 b	1.8 ± 0.2 b*	2.2 ± 0.1 b*	Nd	Nd
4	0.5 ± 0.5 C*	3.2 ± 0.2 B	2.0 ± 0.1 B*	Nd	Nd	1.2 ± 0.1 c*	2.8 ± 0.1 b	10.8 ± 0.2 a*	Nd	Nd
5	4.2 ± 0.2 B	2.0 ± 0.3 B	2.0 ± 0.3 B*	Nd	Nd	3.9 ± 0.2 b	1.6 ± 0.3 b	3.7 ± 0.2 b*	Nd	Nd
6	1.2 ± 0.2 C	2.0 ± 0.2 B	1.8 ± 0.5 B*	Nd	Nd	0.7 ± 0.2 c	1.8 ± 0.3 b	4.2 ± 0.3 b*	Nd	Nd
7	4.3 ± 0.1 C*	1.2 ± 0.2 C	8.8 ± 0.5 A*	Nd	Nd	1.0 ± 0.3 c*	1.7 ± 0.4 b	4.7 ± 0.4 b*	Nd	Nd
8	4.7 ± 0.1 B	1.5 ± 0.1 C	1.0 ± 0.2 B*	Nd	6.0 ± 0.1	4.6 ± 0.5 b	1.6 ± 0.4 b	7.5 ± 0.6 a*	Nd	6.2 ± 0.2
9	11.1 ± 0.2 A	7.6 ± 0.1 A*	1.7 ± 0.1 B	Nd	Nd	10.5 ± 0.5 a	8.7 ± 0.1 a*	2.4 ± 0.4 b	Nd	Nd
10	14.0 ± 0.2 A*	0.5 ± 0.3 C*	1.3 ± 0.2 B*	Nd	Nd	4.7 ± 0.2 b*	1.2 ± 0.2 b*	2.8 ± 0.3 b*	Nd	Nd
11	4.9 ± 0.1 B	1.0 ± 0.2 C	1.0 ± 0.3 B*	Nd	Nd	5.0 ± 0.1 b	1.5 ± 0.1 b	2.2 ± 0.1 b*	Nd	Nd
Range	0.5–14.0	0.4–7.6	1.0–8.9		0.0–6.0	0.7–10.5	0.8–8.7	2.0–10.8		
Mean	5.1	1.9	2.8		0.5	3.8	2.2	4.1		
Mn										
1	8.9 ± 0.3 B*	4.8 ± 0.2 B*	Nd	Nd	Nd	10.5 ± 0.3 b*	7.4 ± 0.1 a*	Nd	Nd	Nd
2	9.0 ± 0.4 B*	4.9 ± 0.2 B*	Nd	Nd	Nd	10.2 ± 0.4 b*	7.2 ± 0.2 a*	Nd	Nd	Nd
3	3.8 ± 0.1 C	0.4 ± 0.1 C	Nd	Nd	Nd	3.4 ± 0.2 c	0.6 ± 0.2 b	Nd	Nd	Nd
4	5.0 ± 0.1 C	Nd	Nd	Nd	Nd	5.4 ± 0.1 c	Nd	Nd	Nd	Nd
5	4.7 ± 0.2 C	Nd	Nd	Nd	Nd	5.0 ± 0.1 c	Nd	Nd	Nd	Nd
6	4.0 ± 0.1 C	Nd	Nd	Nd	Nd	4.8 ± 0.2 c	Nd	Nd	Nd	Nd
7	4.2 ± 0.2 C	Nd	Nd	Nd	Nd	4.0 ± 0.4 c	Nd	Nd	Nd	Nd
8	3.0 ± 0.2 C*	Nd	Nd	Nd	Nd	4.6 ± 0.2 c*	Nd	Nd	Nd	Nd
9	4.0 ± 0.1 C	7.8 ± 0.2 A*	Nd	Nd	Nd	4.6 ± 0.3 c	6.8 ± 0.2 a*	Nd	Nd	Nd
10	15.8 ± 0.1 A	Nd	Nd	Nd	Nd	16.0 ± 0.3 a	Nd	Nd	Nd	Nd
11	9.2 ± 0.2 B	Nd	Nd	Nd	Nd	10.0 ± 0.1 b	Nd	Nd	Nd	Nd
Range	3.0–15.8	0.0–7.8				3.4–16.0	0.0–7.4			
Mean	6.5	1.6				6.7	2.0			
Zn										
1	Nd	3.0 ± 0.1 A	4.5 ± 0.2 B	3.4 ± 0.1 B*	Nd	Nd	3.2 ± 0.1 a	4.0 ± 0.1 b	2.8 ± 0.1 b*	Nd
2	Nd	3.5 ± 0.2 A*	3.6 ± 0.2 B	3.0 ± 0.2 B	Nd	Nd	2.0 ± 0.2 a*	3.2 ± 0.2 b	2.6 ± 0.2 b	Nd
3	Nd	2.6 ± 0.1 A*	2.8 ± 0.1 B	1.2 ± 0.4 C	Nd	Nd	4.7 ± 0.1 a*	3.0 ± 0.1 b	1.6 ± 0.5 c	Nd
4	Nd	2.8 ± 0.2 A*	Nd	11.8 ± 0.1 A	Nd	Nd	3.4 ± 0.2 a*	Nd	10.4 ± 0.4 a	Nd
5	8.0 ± 0.3 A	4.3 ± 0.3 A	Nd	5.0 ± 0.2 B*	Nd	7.8 ± 0.3 a	4.7 ± 0.3 a	Nd	3.6 ± 0.3 b*	Nd
6	Nd	1.2 ± 0.3 B*	Nd	4.8 ± 0.2 B	Nd	Nd	4.0 ± 0.3 a*	Nd	5.6 ± 0.2 b	Nd
7	Nd	1.0 ± 0.4 B*	3.4 ± 0.1 B	1.0 ± 0.3 C*	Nd	Nd	3.7 ± 0.5 a*	4.0	2.8 ± 0.3 b*	Nd
8	9.2 ± 0.2 A	3.4 ± 0.1 A	11.8 ± 0.1 A	5.4 ± 0.4 B*	Nd	9.3 ± 0.1 a	3.2 ± 0.4 a	11.6 ± 0.3 a	6.4 ± 0.1 a*	Nd
9	Nd	3.0 ± 0.2 A	Nd	1.4 ± 0.1 C*	Nd	Nd	3.8 ± 0.5 a	Nd	10.5 ± 0.2 a*	Nd
10	9.2 ± 0.1 A	1.3 ± 0.1 B	Nd	2.9 ± 0.4 C*	Nd	8.6 ± 0.2 a	2.0 ± 0.3 a	Nd	4.7 ± 0.1 b*	Nd
11	Nd	2.8 ± 0.1 A	5.0 ± 0.2 B	3.0 ± 0.2 B	Nd	Nd	2.2 ± 0.1 a	4.6 ± 0.2 b	3.8 ± 0.2 b	Nd
Range	0.0–9.2	1.0–4.3	0.0–11.8	1.0–11.8		0.0–9.3	2.0–4.7	0.0–11.6	1.6–10.5	
Mean	2.4	2.6	2.8	3.9		2.3	3.4	2.4	4.9	

Average ± and standard deviation, n = 3. Different capital letters show differences in organic acid production when comparing endophytic bacteria in 0 M NaCl. Lowercase letters show differences when comparing endophytic bacteria in 2.5 M NaCl. Asterisk (*) shows a significant difference when comparing the same bacteria in 0 or 2.5 M NaCl. In all cases, one-way ANOVA and Tukey´s post hoc test (α = 0.05) were used. Nd = not detected.

**Table 3 plants-13-01626-t003:** Sugar production (µg µL^−1^) by halophilic endophyte bacteria in different ion-solubilization tests under NaCl effect.

	0 M NaCl					2.5 M NaCl				
Bacteria	Ca_3_(PO_4_)_2_	Phytate	K	Mn	Zn	Ca_3_(PO_4_)_2_	Phytate	K	Mn	Zn
	Fructose									
1	55.3 ± 0.4 D*	20.4 ± 1.8 B	129.7 ± 0.1 B*	3.6 ± 0.1 C	3.7 ± 0.4 G	61.3 ± 1.6 d*	21.6 ± 0.8 b	160.5 ± 1.9 b*	3.4 ± 0.1 d	4.0 ± 0.4 h
2	57.5 ± 0.5 D*	2.0 ± 0.6 D	122.8 ± 0.7 B*	4.7 ± 0.2 C	4.7 ± 0.5 G	65.7 ± 2.5 d*	1.8 ± 0.5 c	151.0 ± 2.5 b*	5.0 ± 0.5 d	4.7 ± 0.3 h
3	295.4 ± 1.2 B*	1.7 ± 0.1 D	16.6 ± 0.5 E	16.5 ± 0.4 B	111.1 ± 0.2 D	333.4 ± 1.2 b*	2.8 ± 0.6 c	16.8 ± 0.1 e	18.2 ± 0.2 c	114.8 ± 05 c
4	458.8 ± 0.3 A	2.3 ± 0.5 D	157.5 ± 1.5 A*	32.0 ± 0.2 A	414.0 ± 0.3 A*	459.4 ± 3.0 a	4.7 ± 1.0 c	267.6 ± 0.2 a*	33.4 ± 0.4 a	349.4 ± 0.5 a*
5	Nd	1.7 ± 0.1 D	3.5 ± 1.0 F*	24.7 ± 0.2 B	343.5 ± 0.2 B*	Nd	2.2 ± 0.3 c	8.4 ± 1.0 g*	23.9 ± 0.3 b	83.3 ± 1.0 d*
6	153.1 ± 0.2 C*	93.1 ± 1.0 A	113.3 ± 1.4 B*	18.7 ± 0.5 B	275.3 ± 0.4 C*	308.6 ± 2.0 b*	95.1 ± 0.5 a	128.3 ± 0.1 c*	20.6 ± 0.3 b	282.3 ± 0.5 b*
7	463.8 ± 0.4 A*	2.2 ± 0.3 D	153.1 ± 0.7 A*	35.0 ± 0.3 A*	417.9 ± 0.4 A*	456.2 ± 0.5 a	2.4 ± 0.3 c	264.8 ± 0.1 a*	32.0 ± 0.5 a*	347.4 ± 0.5 a*
8	2.3 ± 0.5 F*	1.9 ± 0.3 D	35.3 ± 0.2 D*	16.4 ± 0.2 B	16.0 ± 0.5 F	6.7 ± 0.6 e	1.5 ± 0.4 c	43.8 ± 0.1 e*	17.4 ± 0.3 c	16.1 ± 0.5 g
9	151.7 ± 0.6 C*	16.2 ± 1.4 B	70.8 ± 0.1 C*	1.9 ± 0.6 D*	3.5 ± 0.6 G	322.8 ± 1.0 b*	18.5 ± 0.9 b	113.1 ± 0.2 d*	3.6 ± 0.2 d*	2.9 ± 0.3 h
10	184.5 ± 0.2 C*	7.0 ± 0.3 C	78.1 ± 0.4 C*	13.1 ± 0.1 B*	38.1 ± 0.2 E	226.7 ± 0.4 c*	7.3 ± 0.1 c	104.0 ± 0.2 d*	16.3 ± 0.5 c*	39.1 ± 0.1 e
11	8.7 ± 0.1 E*	1.8 ± 0.4 D	16.7 ± 0.4 E	9.5 ± 0.3 C	18.8 ± 0.1 F*	19.6 ± 0.6 e*	2.5 ± 0.2 c	18.7 ± 0.1 f*	8.7 ± 0.1 d	25.8 ± 0.1 f*
Range	0.0–463.8	1.7–93.1	3.5–157.5	3.5–417.9	3.5–417.9	0.0–459.4	1.5–95.1	8.4–267.6	3.4–33.4	2.9–349.4
Mean	166.4	13.6	81.5	16.0	149.6	208.4	14.5	116.0	16.5	115.4
	Xylose									
1	Nd	Nd	6.4 ± 0.3 B	Nd	Nd	Nd	Nd	7.4 ± 0.1 b	Nd	Nd
2	Nd	Nd	8.6 ± 0.4 B	Nd	Nd	Nd	Nd	9.6 ± 0.2 b	Nd	Nd
3	Nd	8.5 ± 0.2 A	67.0 ± 0.4 A	Nd	23.2 ± 0.4 A	Nd	8.0 ± 0.4 a	68.0 ± 0.4 a	Nd	24.6 ± 0.6 a
4	Nd	6.4 ± 0.1 AB	65.7 ± 0.1 A	3.6 ± 0.1 A	Nd	Nd	7.8 ± 0.4 ab	67.7 ± 0.5 a	4.8 ± 0.1 a	Nd
5	29.0 ± 1.5 B*	5.7 ± 0.1 B	5.4 ± 0.2 B	Nd	Nd	228.1 ± 2.8 a*	6.4 ± 0.2 b	6.2 ± 0.6 b	Nd	Nd
6	Nd	Nd	8.2 ± 0.1 B	Nd	Nd	Nd	Nd	9.4 ± 0.1 b	Nd	Nd
7	Nd	3.4 ± 0.2 C	70.1 ± 0.3 A	3.8 ± 0.2 A	Nd	Nd	4.2 ± 0.1 b	73.1 ± 0.1 a	4.6 ± 0.3 a	Nd
8	Nd *	Nd	Nd	Nd	Nd	4.4 ± 0.4 b*	Nd	Nd	Nd	Nd
9	Nd	Nd	66.6 ± 0.4 A	Nd	24.6 ± 0.3 A	Nd	Nd	68.6 ± 0.2 a	Nd	25.7 ± 1.0 a
10	229.5 ± 1.8 A*	Nd	Nd	Nd	Nd	Nd *	Nd	Nd	Nd	Nd
11	Nd	Nd	6.4 ± 0.6 B	Nd	Nd	Nd	Nd	7.8 ± 0.3 b	Nd	Nd
Range	0.0–229.5	0.0–6.4	0.0–70.1	0.0–3.8	0.0–24.6	0.0–228.1	0.0–8.0	0.0–73.1	0.0–4.8	0.0–25.7
Mean	23.5	2.1	27.6	0.6	4.3	21.1	2.4	28.8	0.8	4.5

Average ± and standard deviation, n = 3. Different capital letters show differences in sugar production when comparing endophytic bacteria in 0 M NaCl. Different lowercase letters show differences when comparing endophytic bacteria in 2.5 M NaCl. Asterisk (*) shows a significant difference when comparing the same bacteria in 0 or 2.5 M NaCl. In all cases, one-way ANOVA and Tukey´s post hoc test (α = 0.05) were used. Nd = not detected.

**Table 4 plants-13-01626-t004:** Solubilization phosphate-related enzymatic activity in halophilic endophytic bacteria in three NaCl concentrations.

	Alkaline Phosphatase (Nitrophenylphosphate µg mL^−1^ h^−1^)			Acid Phosphatase (Nitrophenylphosphate µg mL^−1^ h^−1^)		
	NaCl (M)			NaCl (M)		
Bacteria	0	1.5	2.5	0	1.5	2.5
1	27.6 ± 2.3 bA	8.4 ± 2.8 aB	3.1 ± 0.9 cC	Nd	52.6 ± 2.7 bA	42.4 ± 1.8 cB
2	34.5 ± 2.7 aA	7.8 ± 2.8 aB	3.1 ± 0.9 cC	Nd	59.7 ± 3.2 aA	48.9 ± 2.4 bB
3	18.0 ± 2.3 cA	8.6 ± 2.3 aB	7.8 ± 1.2 aB	Nd	49.8 ± 2.7 bA	49.3 ± 3.8 bA
4	28.2 ± 2.8 bA	6.2 ± 2.4 aB	4.6 ± 1.9 bB	47.0 ± 1.4 bB	60.3 ± 4.1 aA	57.8 ± 2.9 aA
5	16.4 ± 1.9 cA	9.9 ± 1.9 aB	4.3 ± 1.4 bC	44.9 ± 1.4 bB	62.9 ± 2.8 aA	62.4 ± 2.3 aA
6	27.0 ± 2.4 bA	6.4 ± 1.8 aB	6.8 ± 1.8 bB	27.1 ± 2.3 dC	60.3 ± 4.1 aA	50.7 ± 2.4 bB
7	23.3 ± 1.8 bA	3.6 ± 1.4 bB	3.8 ± 1.9 cB	23.8 ± 3.2 dC	60.3 ± 4.1 aA	42.7 ± 1.9 bB
8	Nd	8.5 ± 1.9 aA	9.2 ± 1.9 aA	Nd	62.9 ± 2.8 aA	42.1 ± 1.4 bB
9	29.8 ± 2.3 bA	9.3 ± 1.9 aB	8.7 ± 1.9 aB	33.2 ± 1.7 cC	63.8 ± 2.8 aA	51.9 ± 2.3 bB
10	25.1 ± 1.8 bA	5.5 ± 1.9 aB	9.6 ± 1.9 aB	51.3 ± 1.2 aC	62.2 ± 2.9 aA	51.3 ± 1.4 bB
11	17.7 ± 2.4 cA	9.6 ± 1.9 aB	7.4 ± 1.4 abB	39.3 ± 5.2 cC	51.3 ± 1.7 bA	44.2 ± 1.8 cB
Range	16.4–34.5	3.6–9.9	3.1–9.5	23.8–51.3	49.8–63.8	42.1–62.4
Mean	22.5	7.9	6.2	26.4	58.7	49.4
	**Phytase** **(Phosphate µg mL^−1^ h^−1^)**					
	**NaCl (M)**					
**Bacteria**	**0**	**1.5**	**2.5**			
1	80.2 ± 1.8 bC	107.1 ± 2.1 bB	124.2 ± 2.8 cA			
2	70.0 ± 1.2 cC	100.2 ± 2.8 bB	122.2 ± 2.4 cA			
3	90.1 ± 2.8 aA	87.6 ± 3.2 cA	94.8 ± 2.5 dA			
4	4.2 ± 2.4 fC	67.4 ± 3.6 dB	117.7 ± 2.8 cA			
5	90.4 ± 2.2 aC	100.1 ± 3.8 bB	144.5 ± 94 bA			
6	70.5 ± 2.6 cC	87.3 ± 3.3 cB	110.2 ± 2.9 cA			
7	70.6 ± 2.1 cC	100.8 ± 3.1 bB	164.7 ± 3.1 aA			
8	Nd	84.7 ± 3.4 cB	144.9 ± 3.5 bA			
9	90.0 ± 2.6 aC	120.9 ± 3.6 aB	154.4 ± 3.4 aA			
10	50.8 ± 2.8 dC	104.2 ± 3.1 bB	140.7 ± 3.1 bA			
11	40.1 ± 2.6 eC	97.1 ± 3.5 bB	147.5 ± 3.2 bA			
Range	4.2–90.4	67.2–120.8	94.8–164.7			
Mean	59.7	96.1	133.0			

Lowercase letters show differences when comparing enzymatic activity between bacteria at their respective NaCl concentrations. Different capital letters show differences when comparing different concentrations of NaCl in the same bacteria. In all cases one-way ANOVA and Tukey´s post hoc test (α = 0.05) were used. Nd = not detected.

**Table 5 plants-13-01626-t005:** Hydrolytic enzyme activity in halophilic endophytic bacteria at three NaCl concentrations.

	Xylanase (µg mL^−1^ h^−1^)			Chitinase (µg mL^−1^ h^−1^)		
	NaCl (M)			NaCl (M)		
Bacteria	0	1.5	2.5	0	1.5	2.5
1	Nd	Nd	Nd	1.62 ± 0.931 aB	6.85 ± 1.021 aA	6.70 ± 0.923 aA
2	Nd	Nd	Nd	1.90 ± 0.161 aB	6.43 ± 0.212 aA	5.48 ± 0.546 aA
3	0.09 ± 0.007 aB	0.76 ± 0.004 bB	3.14 ± 0.005 bA	1.67 ± 0.112 aB	6.92 ± 0.283 aA	6.54 ± 0.283 aA
4	0.14 ± 0.012 aB	0.68 ± 0.005 bB	3.01 ± 0.010 bA	1.24 ± 0.932 aB	6.48 ± 0.827 aA	5.45 ± 1.093 aA
5	0.06 ± 0.008 aC	0.62 ± 0.004 bB	2.14 ± 0.003 bA	1.45 ± 0.836 aB	6.87 ± 0.927 aA	6.30 ± 0.864 aA
6	0.06 ± 0.008 aC	0.67 ± 0.006 bB	2.92 ± 0.005 bA	1.86 ± 0.732 aB	6.56 ± 1.052 aA	6.28 ± 0.834 aA
7	0.02 ± 0.034 aC	0.76 ± 0.005 bB	2.74 ± 0.008 bA	1.90 ± 0.623 aB	5.68 ± 1.024 aA	6.34 ± 0.941 aA
8	Nd	0.79 ± 0.004 bB	2.80 ± 0.005 bA	Nd	6.14 ± 0.962 aA	5.85 ± 0.841 aA
9	0.09 ± 0.005 aC	0.83 ± 0.004 bB	3.22 ± 0.007 bA	1.78 ± 0.924 aB	6.24 ± 0.863 aA	6.35 ± 0.974 aA
10	Nd	Nd	Nd	Nd	Nd	Nd
11	0.08 ± 0.015 aC	1.89 ± 0.010 aB	5.32 ± 0.008 aA	Nd	Nd	Nd
Range	0.02–0.14	0.62–1.89	2.14–5.32	1.24–1.90	5.68–6.92	5.45–6.70
Mean	0.04	0.63	2.29	1.22	5.28	5.02

Lowercase letters show differences when comparing enzymatic activity between bacteria at their respective NaCl concentration. Different capital letters show differences when comparing different concentrations of NaCl in the same bacteria. In all cases one-way ANOVA and Tukey´s post hoc test (α = 0.05) were used. Nd = not detected.

## Data Availability

All information generated in this research is presented.

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
