# Peer review of "NaCl Modifies Biochemical Traits in Bacterial Endophytes Isolated from Halophytes: Towards Salinity Stress Mitigation Using Consortia"

_plants, 2024, doi:10.3390/plants13121626_

Round 1

Reviewer 1 Report

Comments and Suggestions for Authors

I think the manuscript is well-constructed, and the outcomes are generally convincing. However, I believe there is an opportunity to enhance the presentation of data. I recommend relocating a portion of the data to supplementary materials to streamline the main body of the manuscript. This approach would allow for a concentration on essential findings, thereby enhancing clarity and impact. Including seven figures and tables could potentially overwhelm readers, reducing their engagement and comprehension of the material. Therefore, I suggest reducing the figures and tables in the main text. Moreover, I recommend separating the Results and Discussion sections. This structural adjustment could simplify the narrative flow and encourage continued reader engagement.

Furthermore, the results from assessing bacterial hydrolytic enzymatic activity and ACC deaminase production necessitate validation through specific methodologies. An antifungal assay is recommended for the hydrolytic activities to confirm the biological relevance of the observed enzymatic activity. For ACC and IAA production, a Gnotobiotic root elongation assay should be employed to substantiate the effect of ACC deaminase on plant growth promotion under different salinity levels. Such validation will reinforce the overarching conclusions of the study and provide a clear take-home message.

Other comments

Abstract:

Line 25: The numerical details provided alongside the name of the isolated bacterium are somewhat perplexing. Could you please rephrase the sentence to minimize the use of numbers?

Line: 28: Biochemical responses of nutrient solubilize…. Please rewrite the sentence. It is unclear.

The first section of the Results and Discussion section is poorly written. It does not contain a discussion part at all.

Line 990: The standard protocol for IAA production assay is to grow the bacteria in minimal medium culture plus tryptophan. Please revise the experiment.

Comments on the Quality of English Language

Just very minor 

Author Response

1. I think the manuscript is well-constructed, and the outcomes are generally convincing. Response: Thank you for the comment. 

2. However, I believe there is an opportunity to enhance the presentation of data. Response: The authors agree with this reviewer´s point of view: The better the presentation, the wider the number of readers and paper acceptance. Please see all changes made highlighted in red within the new document version.

3. I recommend relocating a portion of the data to supplementary materials to streamline the main body of the manuscript.
This approach would allow for a concentration on essential findings, thereby enhancing clarity and impact. Including seven figures and tables could potentially overwhelm readers, reducing their engagement and comprehension of the material. Therefore, I suggest reducing the figures and tables in the main text. Response:
Some figures and tables were relocated to supplementary information according to suggestions made by the reviewer to concentrate on major findings and enhance clarity and impact. 
Old Tables 1 and 2, and Figures 5 and 6 are now in supplementary information.

4. Moreover, I recommend separating the Results and Discussion sections. This structural adjustment could simplify the narrative flow and encourage continued reader engagement. Response: The results and discussion sections were separated as suggested by the reviewer.

5. Furthermore, the results from assessing bacterial hydrolytic enzymatic activity and ACC deaminase production necessitate validation through specific methodologies. Response: Regarding hydrolytic activity, the procedure employed in identifying xylanase activity (DNS assay for reducing sugar) is the most common method for determining this activity. For the second case, which involved chitinase, a sensitive, reproducible, and suitable method was used in the selection of chitinolytic strains.

The ACC-diaminase test was not analyzed in the present research, it was out of the authors´ scope.

The authors appreciate the reviewer's suggestion; however, the paper is already extensive with several complementary biochemical analyses, providing a big piece of data that may soon be directed toward biological validation of biochemical traits in various environmental conditions and NaCl-sensitive plants. This is mentioned in the corrected version in green highlighted color.

Validation was outside of the authors’ objectives.

6. An antifungal assay is recommended for the hydrolytic activities to confirm the biological relevance of the observed enzymatic activity. Response: The observation regarding the antifungal assay is respected, but its evaluation was not part of the objectives of this work. It remains as a perspective for future research. This is mentioned in the corrected version in green highlighted color.

7. For ACC and IAA production, a Gnotobiotic root elongation assay should be employed to substantiate the effect of ACC deaminase on plant growth promotion under different salinity levels. Such validation will reinforce the overarching conclusions of the study and provide a clear take-home message. Response: The authors respect and value the reviewers´ ACC and IAA production suggestions. As we have previously mentioned, these aspects were outside the authors' objectives. Unfortunately, ACC was not analyzed in the present research. IAA production has been already tested and validated in tomato seedlings; however, this information is part of another committed research. To reinforce the robustness of the document, this is referenced in Section 3.6.

Abstract

8. Line 25: The numerical details provided alongside the name of the isolated bacterium are somewhat perplexing. Could you please rephrase the sentence to minimize the use of numbers? Response: The sentence was rephrased to make it clearer.

Abstract

9. Line: 28: Biochemical responses of nutrient solubilize…. Please rewrite the sentence. It is unclear. Response: The sentence was rewritten as required.

10. The first section of the Results and Discussion section is poorly written. It does not contain a discussion part at all. Response: More discussion sentences have been added to this section.

11. Line 990: The standard protocol for IAA production assay is to grow the bacteria in minimal medium culture plus tryptophan. Please revise the experiment. Response: For this research, the authors followed the protocol used by Bano and Musarrat (2003); who tested the IAA production in LB broth plus tryptophan. Li et al. (2018)* is another recent reference using LB for this test. Because both tests, with and without NaCl, were made in the same medium culture and conditions, the results are reliable.

* Li, M., Guo, R., Yu, F., Chen, X., Zhao, H., Li, H., & Wu, J. (2018). Indole-3-acetic acid biosynthesis pathways in the plant-beneficial bacterium Arthrobacter pascens ZZ21. International Journal of Molecular Sciences, 19(2), 443.                                                                     

Reviewer 2 Report

Comments and Suggestions for Authors

This manuscript reported the identification and characterization of 11 salt-tolerant bacterial endophytes from halophytic plants. Those bacteria could be used as potential tools for mitigation of saline stress in agriculture. This manuscript is well written, and the data were properly analyzed, presented and interpreted. Authors also surveyed the literature well and got their data compared with previous studies, which is very helpful for making comprehensive conclusions and for readers to interpret their data. 

Minor suggestion: in all figure legends, Tukey should be Turkey and it is better to state it like: one-way ANOVA and Turkey’s post hoc test, rather than Turkey alone.  

Author Response

  1. This manuscript reported the identification and characterization of 11 salt-tolerant bacterial endophytes from halophytic plants. Those bacteria could be used as potential tools for mitigation of saline stress in agriculture. This manuscript is well written, and the data were properly analyzed, presented and interpreted. Authors also surveyed the literature well and got their data compared with previous studies, which is very helpful for making comprehensive conclusions and for readers to interpret their data. Response: Thank you for the comment.
  2. In all figure legends, Tukey should be Turkey and it is better to state it like: one-way ANOVA and Turkey’s post hoc test, rather than Turkey alone. Response:  The authors added the suggestion made by the Reviewer 2. One-way ANOVA and Tukey´s post hoc test were added in all figures and tables. 
